# RePlan: Robotic Replanning with Perception and Language Models

## Abstract

Advancements in large language models (LLMs) have demonstrated their potential in facilitating high-level reasoning, logical reasoning and robotics planning. Recently, LLMs have also been able to generate reward functions for low-level robot actions, effectively bridging the interface between high-level planning and low-level robot control. However, the challenge remains that even with syntactically correct plans, robots can still fail to achieve their intended goals. This failure can be attributed to imperfect plans proposed by LLMs or to unforeseeable environmental circumstances that hinder the execution of planned subtasks due to erroneous assumptions about the state of objects. One way to prevent these challenges is to rely on human-provided step-by-step instructions, limiting the autonomy of robotic systems. Vision Language Models (VLMs) have shown remarkable success in tasks such as visual question answering and image captioning. Leveraging the capabilities of VLMs, we present a novel framework called Robotic **Re**planning with **P**erception and **Lan**guage Models that enables real-time replanning capabilities for long-horizon tasks. This framework utilizes the physical grounding provided by a VLM's understanding of the world's state to adapt robot actions when the initial plan fails to achieve the desired goal. We test our approach within four environments containing seven long-horizion tasks. We find that RePlan enables a robot to successfully adapt to unforeseen obstacles while accomplishing open-ended, long-horizon goals, where baseline models cannot. Find more information at https://sites.google.com/view/replan-iclr/home

## 1 Introduction

Designing **embodied agents** to execute **multi-stage, long-horizon** tasks is challenging. Firstly, agents need **manipulation skills** for **physical engagement** with their environment. They also need to be adept at **perceiving** their surrounding environment and **reasoning** on cause-and-effect relationships of their actions on the environment. Moreover, these agents should be able to **plan** and carry out a series of actions that are in line with the main goals they are tasked to accomplish (Wu et al., 2023), with minimal human intervention.

Methods based on rule-driven frameworks like Task and Motion Planning (TAMP) (Garrett et al., 2021) and learning approaches, such as Hierarchical Reinforcement Learning (HRL) and Imitation Learning (IL), have advanced the field of long-horizon planning. Yet, these methods often require extensive domain knowledge, intricate reward engineering, and time-consuming dataset creation efforts (Hussein et al., 2017; Brohan et al., 2022). In contrast, the rise of Large Language Models (LLMs) has shown considerable promise in robot planning (Driess et al., 2023; Brohan et al., 2023a). However, applying LLMs in this area is complex, mainly because LLMs are used for generating open-ended text while robots require constrained code (Singh et al., 2022; Wang et al., 2023a).

Long-horizon, multi-stage task planning requires reasoning over extended periods, which is a challenge for LLMs Wang et al. (2023c). In the context of LLMs, tackling large-scale problems often leads to issues like hallucination or failing to consider important details, rendering their plans ineffective or error-ridden (Bang et al., 2023; Wang et al., 2023b). To address this, prompting schemes like **ReAct** and **Chain-of-Thoughts** distil complex problems into intermediate reasoning steps with the aid of exemplars, facilitating effective reasoning (Yao et al., 2023; Wei et al., 2022). However, these efforts are still constrained by the number of stages the robot can handle. Another strategy to

enhance LLM outputs involves **verify**ing their results, employing techniques such as syntax checking (Skreta et al., 2023; Wang et al., 2023a) and semantics (Rana et al., 2023) verification, as well as simulating task execution (Liu et al., 2022) to provide success/failure feedback to the LLM. In those works, verification enables the ability to plan over multiple steps in complex domains.

Working with multi-step tasks in robotics also involves dealing with uncertainties and changes in the environment. Effectively handling these tasks requires combining task instructions with sensory data, which helps the robot adapt to the changing surroundings. Recent research has shown that combining textual and visual information in RL or IL can improve task performance (Ha et al., 2023; Brohan et al., 2022; 2023b). It allows for ongoing updates based on what the robot observes. Vision-Language Models (VLMs) have been shown to guide robots in interpreting their environment more accurately and generalizing to unseen scenarios. By integrating visual cues with linguistic context, VLMs enable robots to better interpret their surrounding environment (Brohan et al., 2023a).

Furthermore, for achieving generalization and scaling in the deployment of robots across diverse environments, the acquisition of rich low-level skills is essential. Many works have utilized pre-trained and encoded skills, supplying a skill list to LLMs for skill selection (Lin et al., 2023). Recently, there has been a shift towards directly setting up **rewards generated by LLMs** (Yu et al., 2023; Kwon et al., 2023; Xie et al., 2023). This approach, employed in both RL for policy learning (Kwon et al., 2023; Xie et al., 2023) and Model Predictive Control (MPC) (Garcia et al., 1989; Rawlings, 2000) to enhance data efficiency (Miyaoka et al., 2023; Yu et al., 2023), enabling users to more easily guide robot behavior by creating and combining rewards.

This paper aims to address the challenge of multi-stage long-horizon tasks, inheriting key ideas from recent progress in foundation models. We introduce REPLAN, an innovative *zero-shot approach* that harnesses LLMs at multiple levels by iterative re-prompting to serve as a reward generator for robot MPC. Differently from (Yu et al., 2023), our approach is augmented by high-level replanning based on contextual feedback from VLMs and motor control observations. Our approach can solve open-ended problems, such as searching tasks. REPLAN is hierarchical, composed of two planners, one high-level and one low-level. The output of each level is passed through a verifier. Our system operates by taking a natural language goal from the user, which is used to generate high-level plans, followed by the creation of low-level plans to define reward functions for the controller. By utilizing inputs from robot motor signals and raw visual scene feedback, a Perceiver model offers online feedback to the LLM Planner, facilitating plan progress monitoring and replanning.

Our main contribution in handling long-term, multi-stage tasks involves four key aspects: using perceiver models for high-level replanning, creating hierarchical plans with language models, verifying outputs from these language models, and robot behavior through reward generation. By combining these elements, our approach can create effective multi-step plans with significantly less need for human involvement. To test these abilities, we develop new multi-stage planning domains. Across seven long-horizon tasks requiring up to 11 steps on average, our method was able to succeed almost $4\times$ as often as the current leading method.

## 2 RELATED WORK

**Long-horizon Robot Planning.** Addressing long-horizon planning in robotics has been a persistent challenge. Rule-based methods (Mehr et al., 2020; Baier et al., 2009), such as Planning Domain Definition Language (PDDL) (Aeronautiques et al., 1998), attempted to solve task and motion planning sequentially, however, the planned task may not feasible when the domain knowledge is incomplete. The task and motion planning (TAMP) approach (Garrett et al., 2021), addresses this by simultaneously determining symbolic actions and low-level motions. For example, PDDLStream (Garrett et al., 2020), introduces streams to streamline continuous robot planning into a finite PDDL problem while ensuring action feasibility during planning. While these methods excel in *verifying* task and motion plans during planning, their *generalization* to new environments is constrained, particularly when tackling intricate tasks, and they necessitate substantial domain description engineering. In addressing multi-stage planning challenges, many works focus on learning task plans from input task specifications, leveraging reinforcement learning (RL) and imitation learning (IL) techniques. For example, Behavior-1K (Li et al., 2023) employs RL to acquire semantics and physical manipulation skills, often benefiting from classical motion planners and simplifying assumptions. However, it's important to note that these learning-based techniques demand significant domain expertise for re-

ward engineering and rely on large datasets for task learning (Heo et al., 2023). While they adeptly *react* to environmental uncertainties by iteratively updating policies based on observations, their zero-shot generalization across multi-stage tasks remains a persistent challenge.

**Robot Control with Physically Grounded Language Models.** Recent advancements in LLMs have resulted in their adoption in robot planning, leveraging their natural language capabilities and common-sense reasoning for generating robot task and motion plans(Wang et al., 2023b; Xi et al., 2023). Notably, LLMs have been applied to planning multi-stage tasks (Singh et al., 2022; Driess et al., 2023), by utilizing LLMs to improve sample efficiency in reinforcement learning. The Prog-Prompt and Code-As-Policies approaches use code-writing LLMs to generate code for robot policies (Singh et al., 2022; Liang et al., 2022). Language models with a verifier have been used for generating long-horizon tasks in an iterative prompting technique with lower error rates (Skreta et al., 2023), however there is no guarantee that the output task plan can be executed. SayPlan used LLMs to reason over scene graphs and generate plans across large environments, using iterative replanning to ensure scene graph constraints were not violated (Rana et al., 2023). Toward grounding the language belief with visual and motor control feedback, (Ha et al., 2023) employed language guidance for skill learning, using LLMs alongside sampling-based planning and visuomotor policy learning. (Brohan et al., 2023a) proposed a vision-language-action model, co-training it on internet-scale data with other image-related tasks to enhance generalization. Additionally, (Stone et al., 2023) interfaced robot policies with pre-trained vision-language models, enabling interaction with unseen object categories based on language descriptions and localization data. SayCan uses affordances based on surrounding observations to constrain LLM plans to be more feasible given the environment Brohan et al. (2023b). Inner Monologue extends this line of work by incorporating feedback from other sensors while generating plans, including passive scene descriptions, active human feedback to guide the robot, and success detection, enabling it to retry low-level policies if actions failed Huang et al. (2022b) and therefore execute longer-horizon tasks than SayCan. It has observed in other works that binary success indicators are usually not enough to long-horizon correct plans Wang et al. (2023c); Skreta et al. (2023), although the tasks in Inner Monologue focused on retrying low-level policies instead of overcoming environment obstacles. Finally, LLMs have been used to enable open-ended world exploration of embodied agents. In Voyager, an agent is encouraged to explore the world of Minecraft and build a skill library Wang et al. (2023a). Voyager demonstrates that receiving environment feedback from a chat bot as to why tasks cannot be completed improves the likelihood of task completion.

**Language to Reward Shaping.** In contrast to approaches that map natural task descriptions to robot actions and subsequently to rewards, an alternative approach seeks to directly infer rewards from natural language inputs, addressing the challenge of reward engineering (Lin et al., 2022). This language-driven reward-shaping approach has demonstrated utility in various domains, including negotiation (Kwon et al., 2023) and gaming (Goyal et al., 2019), facilitating desired behavior learning through RL. (Mahmoudieh et al., 2022) introduce a visuo-language model that generates robot motion policy reward on goal text and raw pixel observations, in a manner similar to (Radford et al., 2021), enabling zero-shot prompting for unseen scenarios. (Yu et al., 2023) employs an iterative prompting method using a LLM to link user task specifications and robot motion through reward functions. While excelling in motion generation with minimal data, their approach falls short in handling long-horizon multistage tasks and lacks real-time environment feedback, necessitating user intervention for adaptation. (Xie et al., 2023) extended the previous work for robot reward policy refinement by requiring substantial human involvement and Pythonic observation from the environment. Both of these methods struggle with open-ended problems and multi-stage tasks. To mitigate these limitations, our work autonomously performs long-horizon tasks and adapts to execution outcomes by leveraging motor control and raw visual feedback.

## 3 REPLAN: MODEL STRUCTURE AND DETAILS

We present an overview of our method in Figure 1. The input to our system is a goal described in natural language. The goal can be specific (e.g. place kettle on stove) or open-ended (e.g. search for the banana). REPLAN has five modules, which are described below. All prompts used for the modules can be found in Appendix B.

1. a High-Level LLM Planner used for planning, replanning, and reasoning

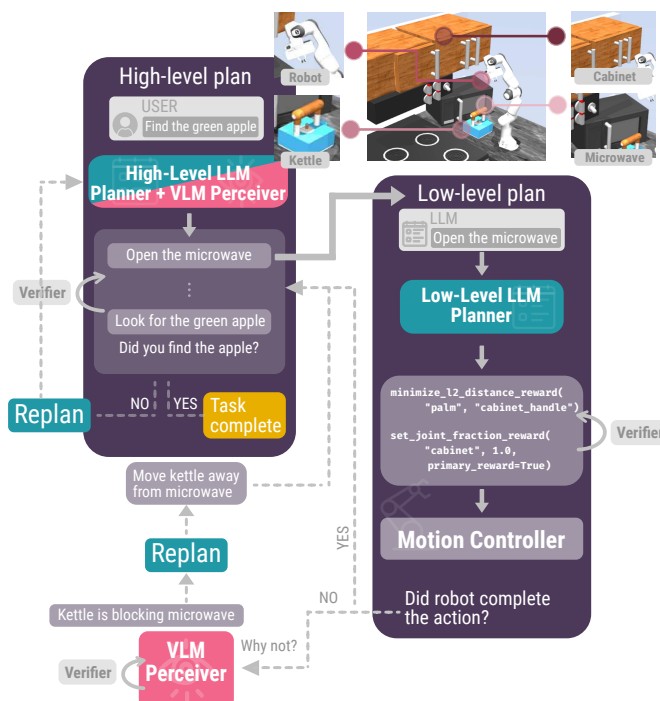

Figure 1: REPLAN overview. It consists of five modules: a High-Level LLM Planner, a VLM Perceiver, a Low-Level LLM Planner for low-level reward generation, a motion controller with motor-control feedback, and an LLM Verifier. The robot's task is to locate a green apple hidden in the microwave. Initially, the VLM provides a list of verified objects to the high-level planner. The LLM generates verified high-level plans for the robot's execution. These high-level plans are sequentially passed (e.g., "open the microwave") to the low-level planner to generate verified low-level plan descriptions and corresponding rewards. These rewards are subsequently conveyed to the MPC controller for robot execution. Upon successful completion of the low-level plan (determined by motor control signals), it notifies the high-level planner. If a failure occurs, we query the VLM to obtain textual information about it, which helps REPLAN effectively resolve the task.

2. a VLM Perceiver used for physically-grounded reasoning

3. a Low-Level LLM Planner used for converting high-level tasks to low-level rewards

4. a Motion Controller to instruct the robot on low-level actions

5. a LLM Verifier to check that the Planner/Perceiver is correct and fixes them if applicable

## 3.1 HIGH-LEVEL LLM PLANNER

Inspired the ability of LLMs to generate actionable steps from high-level tasks (Huang et al., 2022a), we employ a High-Level Planner to take in as input a user-specified task and return a list of subtasks on how a robot should accomplish the task. We use a prompting scheme similar to ReAct Yao et al. (2023) for generating subtasks. The benefit of using a High-Level Planner is that there are no restrictions on the abstraction level of the user input. The user input can be a specific task (e.g. "Place my keys on the counter"), or an open-ended task where the procedure requires exploration (e.g. "Find my keys"). This is because the Planner can propose a procedure, even if it doesn't know the exact answer a priori. If the robot completes the sequence of actions proposed by the Planner and the overall goal is still not accomplished, the High-Level Planner can propose a new procedure. The High-Level Planner can also utilize the past recommended procedures to prevent the redundancy of having the agent perform the same tasks repetitively (Skreta et al., 2023). The High-Level Planner is also used to incorporate feedback from perception models when generating high-level plans. This is important because the Planner should not generate plans that are not aligned with the physical state of the world, or should be able to replan if there are any obstacles.

## 3.2 VLM PERCEIVER

While LLMs have demonstrated powerful reasoning skills over text, they lack grounding in the physical world (Liu et al., 2022). This means that while LLMs can generate plans that sound reasonable, they may fail to account for any obstacles or uncertainties that are present because they cannot perceive the environment. At the same time, while VLMs offer physical grounding to text queries, their language generation capabilities are not as strong as those of LLMs. Considering this, we use the

High-Level Planner to decide what it wants to query from the Perceiver, and then the Planner incorporates feedback from the Perceiver when it needs to know about the object states or replan because the robot failed to do an action. The High-Level Planner decides on specific, simple questions to ask the Perceiver (see Figures B.14-B.16) and samples multiple answers before consolidating them into a summary observation that is consistent with the state of the environment (see Figures B.9-B.12).

### 3.3 Low-Level LLM Planner

Recently, it has been demonstrated that LLMs are capable of producing low-level plans that enable robot motion control(Yu et al., 2023; Xie et al., 2023). This is exciting because it bridges the gap between high-level, human-specified goals and robot actions in a zero-shot manner (without the need for extensive training datasets). However, while previous works are able to generate low-level robot actions for a concrete task (e.g. "Open the drawer"), we find that they fail when asked to generate plans for long-horizon, open-ended tasks. Thus, we utilize the High-Level Planner to generate concrete subtasks from a high-level goal, which is then passed to the Low-Level Planner to generate the corresponding low-level actions. Our Low-Level Planner uses the same Reward Translator as in (Yu et al., 2023) which we have found works well. The Low-Level Planner works in two stages. First, it generates a motion plan from a user-specified input. The motion plan is a natural language description of the actions a robot should do to achieve the goal. Then, the motion plan is then translated to reward functions, which serve as a representation of the desired robot motion. These reward functions are then passed to the Motion Controller.

### 3.4 Motion Controller

The Motion Controller receives reward functions and instructs the robot on what actions to do in order to satisfy those functions. For motion control, we use MuJoCo MPC (MJPC), an open-source real-time predictive controller, implemented on MuJoCo physics (Howell et al., 2022). Given the initial condition $x_0$, the control problem is defined as:

$$\underset{x_{1:T}, u_{1:T}}{\text{minimize}} \quad \sum_{t=0}^{T} c(x_t, u_t), \quad \text{subject to} \quad x_{t+1} = f(x_t, u_t),$$

where $x_t$ and $u_t$ are the state and control signals at time step $t$, and the mapping $f$ is the transition dynamics. The goal is to minimize the cost function $c(.)$ along the trajectory from the current time step to the horizon $T$. We define the $M$ output rewards provided by the low-level planner as the negative of the cost function, i.e., $c(x_t, u_t) = -\sum_{i=1}^{M} w_i \, r_i(x_t, u_t, \phi_i)$, where $\phi_i$ and $w_i$ are the $i$'th reward parameters and weight. To solve the optimization problem, the predictive sampling implementation is used in our work (Howell et al., 2022).

A subtask can have more than one reward function, the Low-Level Planner also reasons about which reward function actually determines the success state of the action. For example, for the subtask: `Move the kettle away from the microwave`, the Planner generates:

```
minimize_l2_distance_reward("palm", "kettle")
maximize_l2_distance_reward("kettle", "microwave_handle")
```

The Planner is able to reason that once the function `maximize_l2_distance_reward ("kettle", "microwave_handle")` has been satisfied, the kettle has been moved from the microwave. Thus, the Planner correctly labels this as the primary reward function.

### 3.5 LLM Verifier

LLM outputs can contain errors and are sometimes nondeterministic. One way to increase consistency and reliable is to verify the outputs. We do that by taking plans produced by both the High-Level and Low-Level Planners and asking the Verifier to verify that every step proposed is necessary to achieve the goal. For the Low-Level Planner, the Verifier is used to determine whether each step in the generated motion plan is useful for completing the subtask. This eliminates any unrelated actions that the Low-Level Planner. Motion control is a hard problem, eliminating any unnecessary actions increases the chances of the subtask being completed.

VLMs are prone to hallucination (Dai et al., 2023), and so they are used in a very constrained manner, and so the Verifier is also used to correct any observations made by the Perceiver based on objects that it knows exist in the environment. For example, VLMs can identify an object using different synonyms, which is easy for humans to understand. However robots require instructions that adhere to strict syntax rules. Thus, the Verifier is useful in making sure that plans generated using perceived objects are in line with the object names that the robot can interact with.

---

**Algorithm 1** REPLAN

---

**Input:** user goal prompt $g$, scene image observation $o_{img}$
**Output:** motion controller rewards and their weights $\boldsymbol{r} = \{r_1, \ldots, r_M, w_1, \ldots, w_M\}$, success/failure of planner $done$

1: $memory = \emptyset, done = \text{False}$
2: $s_{scene} = \text{PERCEIVERSCAN}(o_{img})$ ▷ scene state, Prompt B.1
3: **while** $i < num\_retries$ **do**
4:     $subtasks = \text{HIGHLEVELPLANNER}(g, s_{scene}, memory)$ ▷ Prompt B.2
5:     **for all** $subtask, action\_type \in subtasks$ **do**
6:         **if** $action\_type = \text{MPC}$ **then**
7:             $\boldsymbol{r} = \text{LOWLEVELPLANNER}(subtask)$ ▷ Prompts B.4-B.8
8:             $e, o_{img} = \text{MOTIONCONTROLLER}(robot, \boldsymbol{r})$ ▷ motor control error
9:             **if** $e \neq \emptyset$ **then**
10:                 $cause = \text{PERCEIVERDIAGNOSE}(o_{img}, e, subtask)$ ▷ Prompts B.9-B.12
11:                 **if** $cause \neq \emptyset$ **then**
12:                     $subtasks_{replan} = \text{HIGHLEVELPLANNER}(subtask,$
                                        $s_{scene}, cause)$ ▷ Prompt B.13
13:                     $subtasks = subtasks_{replan}$
14:                     $\text{RESTART}(subtasks, \text{line:6})$
15:         **else if** $action\_type = \text{VLM}$ **then**
16:             $q = \text{HIGHLEVELPLANNER}(subtask)$ ▷ Prompts B.14 - B.15
17:             $s_{scene} \leftarrow s_{scene} \cup \text{PERCEIVERACT}(o_{img}, q, subtask)$ ▷ Prompt B.16
18:             $\text{RESTART}(subtask, \text{line:6})$ ▷ Prompt B.17
19:         $done = \text{TASKCOMPLETION}(g)$
20:         **if** $done$ **then**
21:             **break**
22:     $memory \leftarrow memory \cup subtasks$ ▷ Prompt B.18

---

Algorithm 1 describes our model. A user inputs a goal, which is fed into the High-Level Planner along with the scene state from the Perceiver. The Planner generates subtasks, which are then either fielded to the Low-Lever Planner to generate MPC reward functions or the Perceiver to get information about the state. If MPC is required, the success of the action is returned from the Motion Controller. If an error is returned, the Perceiver is asked if there was a cause. If a cause is returned, the Planner is asked to replan to overcome the obstacles. If the cause is unknown or the robot does not achieve the goal after having done the subtasks, the Planner generates a new procedure.

## 4 EXPERIMENTS

### 4.1 ENVIRONMENTS AND TASKS

In order to assess the long-term planning, as well as the logical and low-level planning capabilities of our system, we devised four distinct environments with seven total tasks for testing. The readers can refer to Figure A.1 or our website for visualizations.

#### 4.1.1 WOODEN CABINET SCENE

A room where there is a yellow cube placed on the floor beside a wooden cabinet. There is a red bar holding the handles of the wooden cabinet closed. The doors of the cabinet cannot be opened without removing the bar. We implemented three tasks in this scene:

**Task 1: Place the yellow cube in the wooden cabinet (easy mode).** The robot must pick up the yellow cube and place it in the cabinet, which is open. This is a simple 1-step task that evaluates the Low-Level Planner for single motion planning.

**Task 2: Place the yellow cube in the wooden cabinet (hard mode).** This task is the same as the previous task, but now the wooden cabinet doors are closed. This task requires two steps: 1) opening the door and 2) placing the cube inside the cabinet. The High-Level Planner is assessed to plan a sequence of actions and pass them to the Low-Level Planner for motion generation.

**Task 3: Place the yellow cube inside the wooden cabinet (expert mode).** The challenge with this task is that the robot must identify that it cannot open the wooden cabinet because there is a bar across the handles of the door. After removing the bar, the robot can open the cabinet door and finish the task. This task is challenging because it requires vision to identify that the door cannot be opened, followed by replanning to remove the item blocking the door.

### 4.1.2 KITCHEN ENVIRONMENT SCENE

A kitchen that contains a cabinet with two doors, a microwave, a kettle, and a green apple.

**Task 4: Find the green apple.**   A green apple was hidden in the microwave is not visible to the robot at the start of the scene. The robot must search for the apple in the kitchen. There is an additional challenge where the kettle is blocking the microwave door from being opened, and so to open the door, the robot must first remove the kettle. Same as Task 3, Task 4 also requires both vision and replanning to solve the task, but it has an additional challenge because the goal requires open-ended exploration (it is unclear where the apple is), which requires replanning at a high level.

### 4.1.3 WOODEN CABINET AND LEVER SCENE

A room containing a wooden cabinet, a blue block, and a lever that controls whether the cabinet door is locked or unlocked.

**Task 5: Remove the blue cube from the cabinet.**   Just as with tasks 1-3, this task requires the robot to open a cabinet. There is no physical obstruction preventing the cabinet from being opened; however, the cabinet is locked. The cabinet becomes unlocked once a lever close to the cabinet is pulled. Thus, after (unsuccessfully) trying to open the cabinet door, the robot must reason that it should pull the lever first and then open the door.

### 4.1.4 COLOURED CUBES SCENE

A room containing a small red crate and two cubes (one is yellow and the other is red).

**Task 6: Place the cube with the same colour as the crate on the crate.**   In this task, the robot has to identify the cube with the same colour as the crate and place it on the crate.

**Task 7: Blocking cube.**   The robot is given the colour of a cube it must put on the crate. However, there is already a cube on the crate with a different colour and the crate can only hold one cube at a time. The robot must remove the cube that is already on the crate before placing the target one.

These environments were implemented in MuJoCo (Todorov et al., 2012). We used furniture_sim [1] for the assets. We used the MuJoCo MPC (Howell et al., 2022) to generate the motion control.

## 4.2 EXPERIMENT SETUP

We evaluate our framework using a dexterous robot manipulator simulated using MuJoCo MPC (MJPC) (Howell et al., 2022). As in (Yu et al., 2023), we use a 7 DoF Franka Emika arm.

For the LLM modules in REPLAN, we use OpenAI GPT `gpt-4`[2]. For the VLM Perceiver, we used `Qwen-VL-Chat-7B` (Bai et al., 2023) (except for Task 7, where we used GPT-4V due to hardware constraints). We show the performance of state-of-the-art VLMs on Perceiver tasks in Appendix C.2. We found that from the open-source models we tested, Qwen has the best object reasoning skills; however, its object recognition capabilities improve when we first segment the scene using a segmentation model (we use Segment Anything  Kirillov et al. (2023)). GPT-4V had the best performance overall across object recognition and object reasoning, but there is a strict rate limit on its API.

## 4.3 BASELINES AND ABLATIONS

We compare our method to the Language to Rewards (Yu et al., 2023) framework, a one-shot, in-context learning agent. Language to Rewards uses a Reward Translator to translate a high-level goal (such as "open the drawer") to low-level reward functions that are used by a Motion Controller to instruct a robot on what to do. While Language to Rewards does not utilize a VLM to perceive the scene, we give it access to the same objects as our model identifies at the start of the scene. We also

---

[1]https://github.com/vikashplus/furniture_sim/
[2]https://openai.com/research/gpt-4

Table 1: Number of times that the models completed Tasks 1-7 out of 10 runs. Average completion rates across 7 tasks are listed in the last column.

| Model | Task 1 | Task 2 | Task 3 | Task 4 | Task 5 | Task 6 | Task 7 | Average |
|---|---|---|---|---|---|---|---|---|
| REPLAN [full] | **100%** | **100%** | **60%** | **80%** | **100%** | 90% | **90%** | **88.6%** |
| REPLAN [no Verifier] | 80% | 80% | 20% | 60% | 100% | 100% | 80% | 74.3% |
| REPLAN [no Perceiver] | 90% | 90% | 30% | 20% | 50% | 20% | 0% | 42.9% |
| REPLAN [no Replan] | 80% | 70% | 0% | 0% | 0% | 80% | 10% | 34.3% |
| Language to Rewards (Yu et al., 2023) | 90% | 20% | 0% | 0% | 0% | 50% | 10% | 24.3% |

show sample plans using PDDL and PDDLStream in Appendix E for Tasks 1-3 and GPT-4V for Tasks 3 and 6 in Appendix D.

Finally, to demonstrate the importance of all the modules in our pipeline, we do an ablation study on how well the robot can perform each task without each module. We systematically remove the following modules: VLM Perceiver, LLM Verifier, and replanning of High-Level Planner.

## 4.4 RESULTS

We report the success rates of all models on their ability to complete Tasks 1-7 in Table 1. We report the percentage of successful runs (ten runs were done for each task). The number of actions performed by the robot in each task is shown in Figure C.1. On average, most tasks require 7 to 11 steps, with some runs using up to 17 (all without human intervention). We point out that across the seven tasks with ten trials, we run a total of **401** MPC actions and **101** VLM actions, exceeding the evaluations in previous methods such as Language to Rewards, which has 170 MPC actions.

Overall, REPLAN achieves a $3.6\times$ improvement over Language to Rewards. From the seven tasks, Language to Rewards only shows non-negligible success rates in Task 1, which is a single-motion task, and Task 6, where the algorithm can guess the correct cube to move, resulting in a $50\%$ success rate. Meanwhile, REPLAN achieves at least $60\%$ completion rate in all tested tasks, with a close to $90\%$ average completion rate. REPLAN performs the worst in Task 3, which we conjecture to be a result of both the difficulty of removing the bar from the handles and that in certain situations the target block can get stuck in irrecoverable positions. We include some additional error cases in Appendix C.3. Furthermore, we find that PDDL and naive GPT-4V are unable to solve these tasks out-of-the-box. PDDL requires human-provided ground truth information (for example, that the door is blocked) E. PDDLStream is able to solve the problem eventually, but in general, it requires much longer to solve the problem. When we prompt GPT-4V naively to create a plan for solving some tasks, it is unable to naturally identify obstacles or object states in the scene D. This indicates that constrained prompting is required for VLMs to provide useful feedback.

To evaluate the significance of the Verifier, Perceiver, and Replan modules, we notice that REPLAN achieves $+14.3\%$, $+45.7\%$, and $+54.3\%$ improvement compared with removing the three modules respectively. We also notice that removing each of the modules impact the performance more in more complex tasks.

Noticeably, even in simpler tasks such as Task 1 and 2, REPLAN still demonstrates superior performances compared to other tested variants even though replanning is not necessary in the ideal scenario. This shows REPLAN's ability to act as a safe-net mechanism in case of unexpected errors through using a Verifier, improving the overall consistency.

We show a roll-out of the robot completing Tasks 3 and 4 in Figure 2. For Task 4, the High-Level Planner first instructs the robot to look inside the cabinet. The Perceiver informs the Planner that the apple is not in the cabinet. The Planner then instructs the robots to look inside the microwave. However, the motion control fails. The Perceiver is again called and informs the Planner that there is a kettle in front of the microwave. The Planner replans the task, instructing the robot to first move the kettle out of the way. The robot then resumes with the remaining subtasks. It opens the microwave and the Perceiver informs the Planner that the green apple is there and the task is complete.

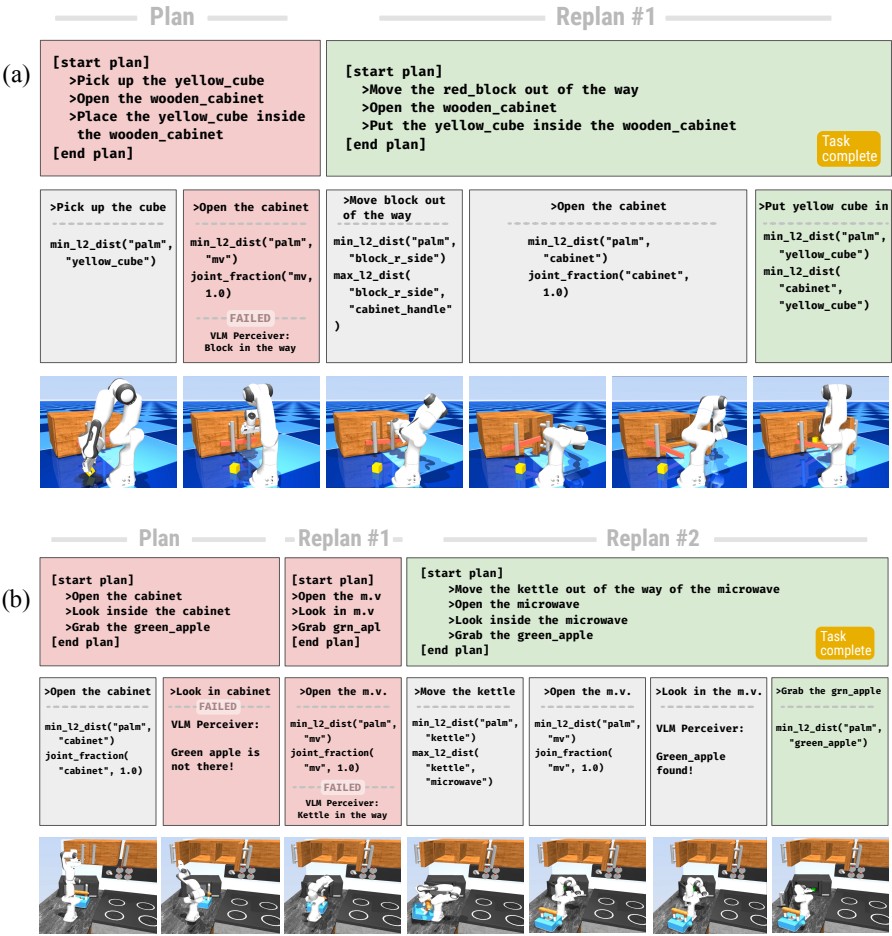

Figure 2: Roll-out of robot solving (a) Task 3 and (b) Task 4. The high-level plan is shown in the top row. The second row shows each subtask and the corresponding reward functions generated by the Low-Level Planner, as well as Perceiver feedback. If the subtask fails, its box is colored in red. If the plan is completed and the goal is achieved, its box is green.

## 5 DISCUSSION, LIMITATIONS, AND CONCLUSIONS

One limitation of our method is its reliance on VLM's understanding and interpretation of spatial states. If the VLM cannot accurately recognize an object or interpret the reason why a task is incomplete, it may lead to the inaccurate processing of the robotic task. However, we find that by using an LLM to probe specific questions, sampling multiple answers, and generating a summary consistent with the LLM's knowledge of the scene, REPLAN is able to utilize the VLM to accurately reason about the scene state. Moreover, there sometimes exists a communication disconnect between LLM and MPC. Currently, the LLM receives task failure feedback as a floating point number, without the ability to discern the specific reasons behind the failure (e.g. the cube was pushed to irretrievable places). Consequently, a failure attributed to the controller could be incorrectly interpreted as the task being undoable. This misinterpretation can lead to unnecessary task abandonment or repeated attempts, resulting in a waste of time and resources. However, we also find that we were able to make MPC reward generation more robust by using a Verifier to eliminate any reward functions that were not essential to completing the task. In essence, we found that using a Verifier at multiple stages in our workflow was essential in improving the success rate of long-horizon task exection.

In summary, our paper introduces REPLAN, a robust solution for multi-stage planning, utilizing the power of LLMs for plan generation and VLMs for insightful feedback. Our multi-level planning approach, coupled with step-wise verification and replanning demonstrates promising results in addressing multi-stage tasks.

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

# A  ENVIRONMENT DETAILS

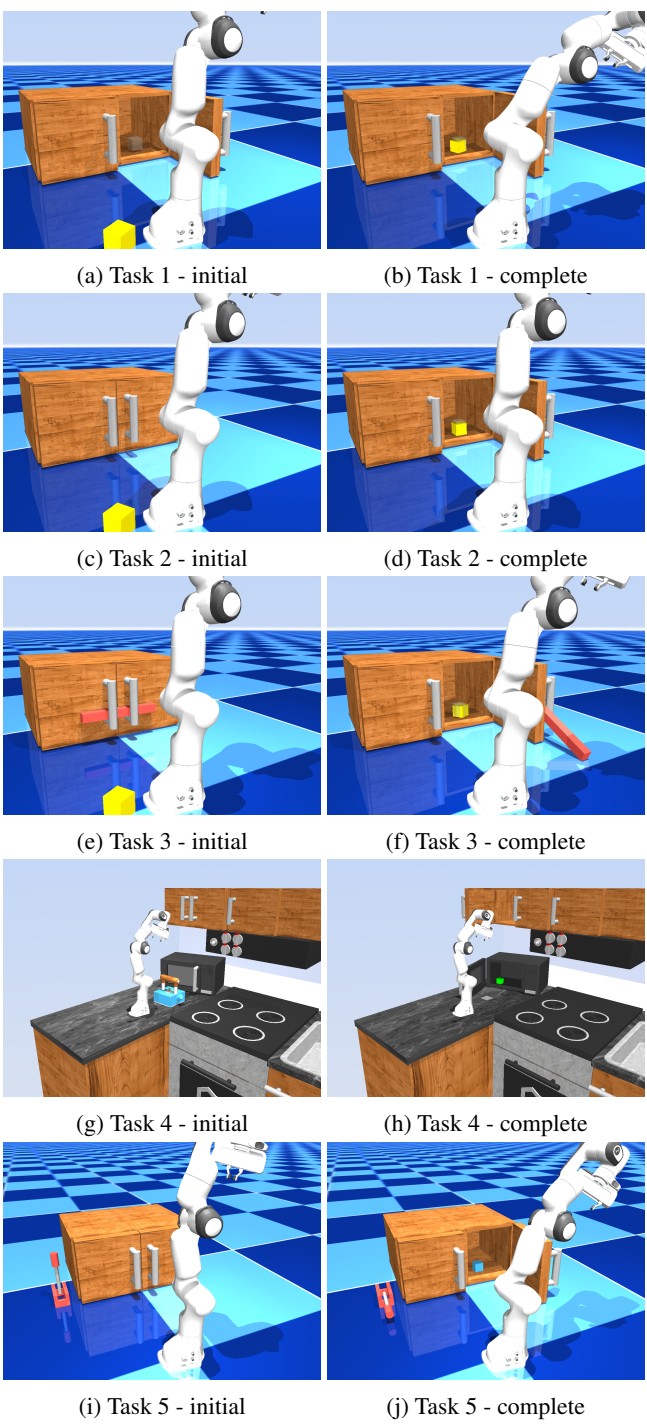

(a) Task 1 - initial

(b) Task 1 - complete

(c) Task 2 - initial

(d) Task 2 - complete

(e) Task 3 - initial

(f) Task 3 - complete

(g) Task 4 - initial

(h) Task 4 - complete

(i) Task 5 - initial

(j) Task 5 - complete

Figure A.1: Initial and final scenes of the tested environments.

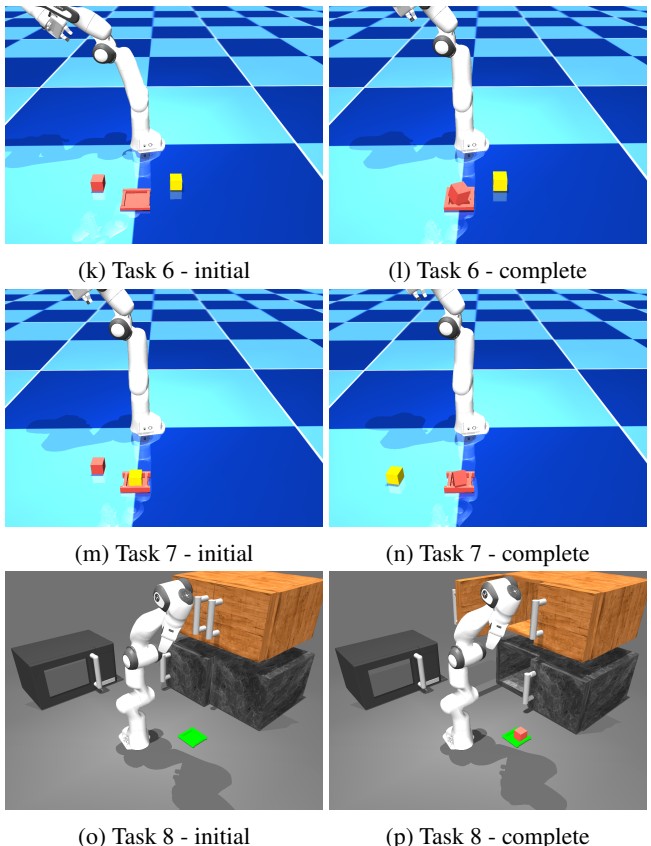

(k) Task 6 - initial        (l) Task 6 - complete

(m) Task 7 - initial        (n) Task 7 - complete

(o) Task 8 - initial        (p) Task 8 - complete

Figure A.1: Initial and final scenes of the tested environments.

The instructions we use for each task are listed below:

| *Environment* | *Instruction* |
|---|---|
| Cabinet (Task 1-3) | `move the yellow_cube to target_position inside the wooden_cabinet` |
| Kitchen (Task 4) | `find the green_cube` |
| Lever (Task 5) | `find the blue_cube` |
| Color (Task 6) | `place the cube with the same color as the crate on the crate` |
| Block (Task 7) | `place the red cube on the crate` |
| Sensor (Task 8) | `open the stone_cabinet. The weight sensor lock can be unlocked by putting the red_cube on it.` |

## B  PROMPTS

Below we show all the prompts we use for the Planners, Perceiver, and Verifier. We show the prompts as the robot would receive them while excuting a task. The prompts are coloured according to the module it comes from – LLM Planners: blue, VLM Perceiver: pink, Verifier: gray.

```
Do you see a(n) {0}?
```

Figure B.1: VLM prompt for perceiving objects in the environment. {0} is an object that the robot knows how to interact with. The VLM is prompted with the list of objects the robot knows how to interact with. If the VLM replies with "yes", that object is added to a list of observed objects.

```
A stationary robot arm is in a location where it sees the following list of objects:

{0}

The robot has the following goal: {1}

Propose high-level, abstract subtasks of what the robot needs to do to {1}. The plan can
 only use one object.

For example, if the goal is to find a fork, one plan might be:

<thought>To find the fork, I will start by looking inside the drawer.</thought>
[start plan]
    >Open the drawer
    >Look inside the drawer
    >Grab the fork
[end plan]

Rules:
1. You have access to the following objects: {0}. Do not create new objects.
2. Generate a plan that interacts with only one object from the list at a time. Keep it
as short as possible. Most plans should be under 5 steps.
3. Assume that every action is completed successfully.
4. Assume the first thing you try works.
5. Your plan should only propose one way of accomplishing the task.
6. The robot only has one arm and it cannot hold two things at a time. Remember that
when you are deciding on the order of actions.
7. Enclose your thought process with a single pair of tag <thought> and </thought>
8. Enclose your plan with the a single pair of tag [start plan] and [end plan]

{2}
```

Figure B.2: LLM prompt for generating high-level task plans. {0} is the list of objects the robot can see (for example: [cabinet, blue kettle, microwave], {1} is the overall task goal (for example: find the green apple), {2} are previous plans that were attempted but failed (see B.18).

```
A robot was asked to do this action:
    > {0}
If the central verb is related to vision, answer yes.
```

Figure B.3: LLM prompt to determine whether the action that the Planner asked the robot to do involves vision or not. If no, then the Planner is called to generate MPC reward functions (see B.4-B.13). If yes, the Perceiver is called (see B.14-B.17). Examples for {0}: "Compare the color of the left cube with the crate", "Open the microwave".

```
We have a stationary robot arm and we want you to help plan how it should move to
perform tasks using the following template:
[start of description]
The manipulator's palm should move close to {{CHOICE: {0}}}.{1}{2}
[end of description]
Rules:
0. You cannot use one line twice!!!!
1. If you see phrases like [NUM: default_value], replace the entire phrase with a
numerical value.
2. If you see phrases like {{CHOICE: choice1, choice2, ...}}, it means you should
replace the entire phrase with one of the choices listed.
3. If you see [optional], it means you only add that line if necessary for the task,
otherwise remove that line.
4. The environment contains {0}. Do not invent new objects not listed here.
5. I will tell you a behavior/skill/task that I want the manipulator to perform and you
will provide the full plan, even if you may only need to change a few lines. Always
start the description with [start of description] and end it with [end of description].
6. You can assume that the robot is capable of doing anything, even for the most
challenging task.
7. Your plan should be as close to the provided template as possible. Do not include
additional details.
8. Your plan should be as concise as possible. Do not include or make up unncessary
tasks.
9. Each object can only be close to or far from one thing.

This is the entire procedure:
{4}

These are the observations we have made so far:
{5}

Create a plan for the following action:
    > {6}
```

Figure B.4: LLM prompt to determine the low-level motion plan for the robot. {0} is the list of objects the robot can interact with. {1} and {2} are modifiers, depending on what type of motion is involved. The Planner is asked to determine whether motion is involved (see B.5. If yes, then {1} becomes: `object1={{CHOICE: {0}}}` should be `{{CHOICE: close to, far from}}` `object2={{CHOICE: {0}}}`. If no, then {1} becomes `[optional]` `object1={{CHOICE: {0}}}` should be close to `object2={{CHOICE: {0}}}`. `[optional]` `object1={{CHOICE: {0}}}` should be far from `object2={{CHOICE: {0}}}`. The modifier {2} is added if there are any joints in the scene that are involved with objects the robot can interact with: `[optional] joint={{CHOICE: {3}}}` needs to be `{{CHOICE: open, closed}}`. (where {3} is the list of joints) (adapted from (Yu et al., 2023)) {4} is the entire plan the robot was given to execute the goal. {5} includes any observations made by the Perceiver (see B.16) using the following format: `Q: <question to Perceiver>, A: <answer from Perceiver>`. {6} is the action the motion plan should be made for.

```
A robot arm has to do this action:
    > {0}
Does this action necessarily involve relocating an object to a different location that
does not involve the robot arm? Answer with yes or no.
```

Figure B.5: LLM prompt to determine if relocation is needed in order to determine the motion plan modifier (see B.4.

```
We have a plan of a robot arm with palm to manipulate objects and we want you to turn
that into the corresponding program with following functions:

    def minimize_l2_distance_reward(name_obj_A, name_obj_B)

where name_obj_A and name_obj_B are selected from {0}. This term sets a reward for
minimizing l2 distance between name_obj_A and name_obj_B so they get closer to each
other. rest_position is the default position for the palm when it's holding in the air.

    def maximize_l2_distance_reward(name_obj_A, name_obj_B, distance=0.5)

This term encourages the orientation of name_obj to be close to the target (specified by
 x_axis_rotation_radians).

    def execute_plan(duration=2)

This function sends the parameters to the robot and execute the plan for "duration"
seconds, default to be 2.

    def set_joint_fraction_reward(name_joint, fraction)

This function sets the joint to a certain value between 0 and 1. 0 means close and 1
means open. name_joint needs to be select from {1}.

    def reset_reward()

This function resets the reward to default values.
Example plan: To perform this task, the manipulator's palm should move close to object1=
faucet_handle. object1 needs to be lifted to a height of 1.0.
This is the first plan for a new task.
Example answer code:

    import numpy as np

    reset_reward()
        # This is a new task so reset reward; otherwise we don't need it
    minimize_l2_distance_reward("palm", "faucet_handle")
    set_joint_fraction_reward("faucet", 1.0)

    execute_plan(4)

Remember:
1. Always format the code in code blocks. In your response execute_plan should be called
 exactly once at the end.
2. Do not invent new functions or classes. The only allowed functions you can call are
the ones listed above. Do not leave unimplemented code blocks in your response.
3. The only allowed library is numpy. Do not import or use any other library.
4. If you are not sure what value to use, just use your best judge. Do not use None for
anything.
5. Do not calculate the position or direction of any object (except for the ones
provided above). Just use a number directly based on your best guess.
6. You do not need to make the robot do extra things not mentioned in the plan such as
stopping the robot.

The action to perform is {2} and the plan is:
{3}
```

Figure B.6: LLM prompt to generate MPC reward functions. {0} is the list of objects the robot can interact with, to which we also append the word "palm" to represent the robot hand. {1} is the list of object joints. {2} is the high-level action the robot needs to perform and {3} is the motion plan generated from B.4. Adapted from (Yu et al., 2023).

```
This is a motion plan generated for a robot:

{0}

This is a reward function generated to complete one step in the motion plan:

{1}

The function minimize_l2_distance_reward() refers to bringing two objects close together
.
The function maximize_l2_distance_reward() refers to moving two objects further apart.
The function set_joint_fraction_reward() refers to opening or closing an object (0 for
closed, 1 for open)
The function set_obj_z_position_reward() specifies the target height of an object.
The function set_obj_orientation_reward() specifies the target rotation of an object.

Which step in the motion plan is the function referring to? Return the step using <step
></step> tags. If it does not refer to any of them, return <step>-1</step>
```

Figure B.7: The Verifier checks that every reward function generated corresponds to a step in the motion plan. If it does not, the function is removed. {0} is the motion plan generated from Figure B.4 and {1} is one of the generated reward functions (they are looped over individually).

```
A stationary robot arm was asked to do the following motion plan to complete the task
'{0}':

{1}

After which step in the motion plan will the task '{0}' be satisfied? First, explain
your thought then answer the step number enclosed with the tag <step> and </step>.
Opening a joint can also mean activating it depending on the context. You must select
one. If you think none of the steps does, select the closest one.
```

Figure B.8: The verifier selects which step in the motion plan is considered to be the most important. The reward function generated for that step in the motion plan becomes labelled as the primary reward function. {0} is the action the robot is currently doing and {1} is the motion plan.

```
A robot is in a simulation environment where it can interact with any object like in the
 real world. The robot would like to {0} but it cannot. Is there something in this scene
 preventing that, other than the robot? Assume the robot can interact with anything.
These are the names of the objects in our scene: {1}
In a simulation, a robot wants to {0} but can't. Is anything else, besides the robot,
blocking it? Check the objects in the scene: {1}.
Robot in a simulation wants to {0}, can't. Something else stopping it? Objects in scene:
 {1}.
A robot can engage with any item. It wants to {0} but can't. Is an object in this scene,
 apart from the robot, hindering it? Objects present: {1}
I would like to {0} but I cannot. Is there something in this scene preventing that,
other than the robot? These are the objects in the scene: {1}
I would like to {0} but I am unable to. Is there something in this scene preventing me
from doing that? Ignore the robot. These are the names of the objects: {1}
```

Figure B.9: If the robot is unable to satisfy the primary reward function, the Perceiver is queried on whether there are any obstacles in the scene. The Perceiver is called once for each question (total of 6). Questions 2-4 and 6 were variations generated for Q1 and Q5, respectively, using ChatGPT.

```
We have access to the following objects in our scene: {0}

You are given a sentence describing an image of the scene, but it may have gotten the
names of the objects wrong. Does this sentence contain objects that are not in our scene
 or get the names of the objects wrong? Start your answer with yes or no.

{1}
```

Figure B.10: For every explanation from the Perceiver, the Verifier is called to determine whether the explanation lists objects that do not exist in the scene. {0} is the list of objects in the scene and {1} is the explanation from the Perceiver. If the Verifier answers with 'yes', the explanation is passed to B.11 for object remapping.

```
We have access to the following objects in our scene: {0}

You are given a sentence describing an image of the scene, but got the names of the
objects wrong. Rewrite this sentence using the closest object(s) in our environment:

{1}

Rules:
You can only use objects in the scene. Use your best judgement.
```

Figure B.11: If the Verifier identifies that the Perceiver explanation contains objects that are not listed in the scene, the Verifier rewrites the explanation ({1}) using the closest objects in our scene ({0}).

```
The stationary robot arm would like to {0} but it cannot. Here are possible reasons why
based on images of the scene:

    {1}

Based on the above explanations, what are the top reason(s) why the robot cannot {0}?
List each reason on a separate line, enclosed with the tag <reason> </reason>. Provide
up to two reasons. Be as succinct as possible. You must not include any reasons related
to the robot, only reasons related to objects in the scene.
```

Figure B.12: The Planner receives all explanations from the Perceiver ({1}, see B.9 -B.11) and summarizes them into key reasons explaining why the robot could not do the action {0}.

```
<Prompt from B.2>

One or more previous attempts failed. Below are the details.
------------------------------ attempt #1 ----------------------------------
This attempt failed when executing '\{0\}'.The plan failed because the robot was not
able to execute this action: '\{1\}'. This was identified as a possible reason the
action failed: '{2}'.
...
------------------------------ attempt #R -----------------------------------
...
-------------------------- end of failed attempts ---------------------------
Reminder to propose a different plan than the above failed attempts.
```

Figure B.13: If the robot does not succeed in performing an action, the Planner is able to replan how the robot does the action by providing the failure reason(s) from the Perceiver from the R failures reasons. Example for {0}: 'Place the red_cube on the crate'. Example for {1}: 'Place the red_cube on the crate (incidentally the action is the same as the overall goal, but it doesn't have to be)'. Example for {2}: 'The most probable reason why the robot cannot place the red_cube on top of the crate is that the yellow cube is currently on top of the crate, which would prevent the robot from doing so.'

```
You are a robot in the process of executing this plan, with the overall goal to '{0}':

{1}

You are currently performing this action: '{2}'. You have access to a perception model
that can answer your questions related to vision.

{3}

What question do you want to ask the perception model in order to get the answer to
'{2}'? You can ask up to two questions. You don't have to ask if the information is
already sufficient. Avoid asking the vision model to compare things. Enclose each of
your questions with the tag <question> </question>.
```

Figure B.14: If the action requires calling the Perceiver, this prompt is used to determine what questions the Planner wants to ask the Perceiver. {0} is the overall goal, {1} is the high-level plan, {2} is the action the robot is currecntly performing, {3} are the observed objects in the scene.

```
What type of question is this asking perception model: '{0}'? Choose your answer from [
OBJECT_PRESENCE, OBJECT_ATTRIBUTE, NEITHER]
```

Figure B.15: Prompt to field what category of question the Planner wants to ask the Perceiver model. {0} is the output from B.14.

```
<Output from B.14. The names of the objects in our scene are: {0}. {1}
```

Figure B.16: Perceiver query on information about the state of objects in the scene from B.14. States are related to object presence or object attributes. Examples of queries: 'Look for the apple in the cabinet', 'Check the color of the crate'.

```
A robot was tasked to do this plan:

{0}

The robot is currently doing this action: '{1}'.

To do the action, the robot asked a perception model the following questions (Q) and got
 the answers (A):

{2}

After receiving this answer, has the robot completed the action '{1}'?  Begin your
answer with yes or no. If your answer begins with no, write the remaining action that
needs to be completed using <Action></Action> tags.
```

Figure B.17: After the Perceiver has provided information, the Planner is asked to determine whether the action is completed. If not, it generates MPC reward functions to finish the action (see B.4-B.18).

```
<Prompt from B.2>

One or more previous attempts failed. Below are the details.
------------------------------- attempt #1 ------------------------------------
The proposed plan was:
<thought>{0}</thought>
[start plan]
{1}
[end plan]
The plan failed because {2}.
...
------------------------------- attempt #P ------------------------------------
...
-------------------------- end of failed attempts --------------------------
Reminder to propose a different plan than the above failed attempts.
```

Figure B.18: If by executing the plan or the replan and the goal is still not accomplished, the Planner is prompted to generate a new plan using the prompt in Figure B.2. The Planner is allowed to generate a new plan $P$ times. before the task is considered undoable.

## C  ADDITIONAL EXPERIMENTS

### C.1  MOTION CONTROLLER

We run additional experiments to test the consistency of our motion controller. Specifically, we pick *4* important motions in our main experiments and run the motion controller *20* times each. The picked motions are:

- *Opening door.* A common motion that opens the cabinet door. This specific tested one is picked from Task 2.
- *Removing bar.* Picked from Task 3, where the robot needs to remove the bar between the door handles.
- *Removing kettle.* Picked from Task 4, where the robot needs to remove the kettle that is obstructing the microwave door.
- *Pulling lever.* Picked from Task 5, where the robot needs to pull the lever switch to unlock the cabinet door.

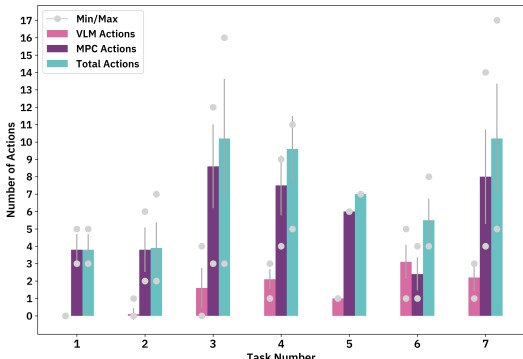

Figure C.1: Number of actions the robot executed in each task averaged over ten runs. Actions requiring the Perceiver are shown in pink while those executed using MPC are shown in purple. Standard deviations are shown using gray bars while the minimum and maximum number of actions are shown using gray dots.

| Motion | Success rate % |
|---|---|
| Opening door | 100% |
| Removing bar | 100% |
| Removing kettle | 80% |
| Pulling lever | 100% |

We would like to comment that these tests are carried out in an ideal setting. During actual planning, multiple facts can affect the motion success rate. For example, the motion planner could generate unnecessary reward functions; and previous steps could change the locations or poses of the objects to interact.

## C.2 VLM ABLATION

We conduct a VLM ablation study for Tasks 3-4 using two open-source models (Qwen-VL-Chat7B (Bai et al., 2023) and Llava-1.5-7B (Liu et al., 2023)) and GPT-4V[3]. The results are shown in Table 2. We measure (a) their ability to recognize objects, reported as a percentage of the number of object successfully detected in the scene (column 2), (b) their ability to reason about obstacles in the scene, reported as a percentage of times the correct reason was identified out of all the prompts it was given in Figure B.9, and (c) the ability for the LLM to consilidate the outputs in (b) into a single summary reason of why the robot cannot perform an action given its knowledge about the scene. We found that Qwen somtimes struggled with object detection of smaller objects, and so we coupled it with Segment Anything Model (SAM) Kirillov et al. (2023) to first segment the objects in the scene. We found that all models did well with object recognition (except for Qwen when not used with SAM). For object reasoning, Qwen + SAM was able to get the correct scene error in 50-67% of the prompts it was given, and then the LLM was able to summarize the prompts to generate the correct error reason overall. The reason the LLM was able to do this despite the VLM not giving perfect answers was that the remaining VLM answers pertained where the robot was located or a general comment about the objects in the scene. Llava tended to reply that it was unable to reason because the scene was a simulation and not real life. GPT-4V had the best overall performance in all categories, but API calls to it are still restricted. All ablations were repeated over 5 runs.

## C.3 ERROR CASES

Here we provide three error cases of REPLAN and their analyses.

**Case 1 (from Task 3, Figure C.2a)** The robot tried to open the cabinet door but failed and the Perceiver gave a correct diagnosis to remove the bar from the handle. However, when generating the reward functions to remove the bar, the LLM selected the wrong primary reward function, as demonstrated below:

---

[3]https://openai.com/research/gpt-4

| Scenarios | Models | | | |
|---|---|---|---|---|
| | Qwen + SAM | Qwen | Llava | GPT-4V |
| VLM object detection | 100% | 66% | 100% | 100% |
| VLM Reasoning | 67% | 0% | 23% | 100% |
| LLM summarization and consistency step | 100% | 0% | 100% | 100% |

(a) Task 3

| Scenarios | Models | | | |
|---|---|---|---|---|
| | Qwen + SAM | Qwen | Llava | GPT-4V |
| VLM object detection | 100% | 100% | 100% | 100% |
| VLM Reasoning | 50% | 66% | 40% | 83% |
| LLM summarization and consistency step | 100% | 100% | 20% | 100% |

(b) Task 4

Table 2: VLM ablation study.

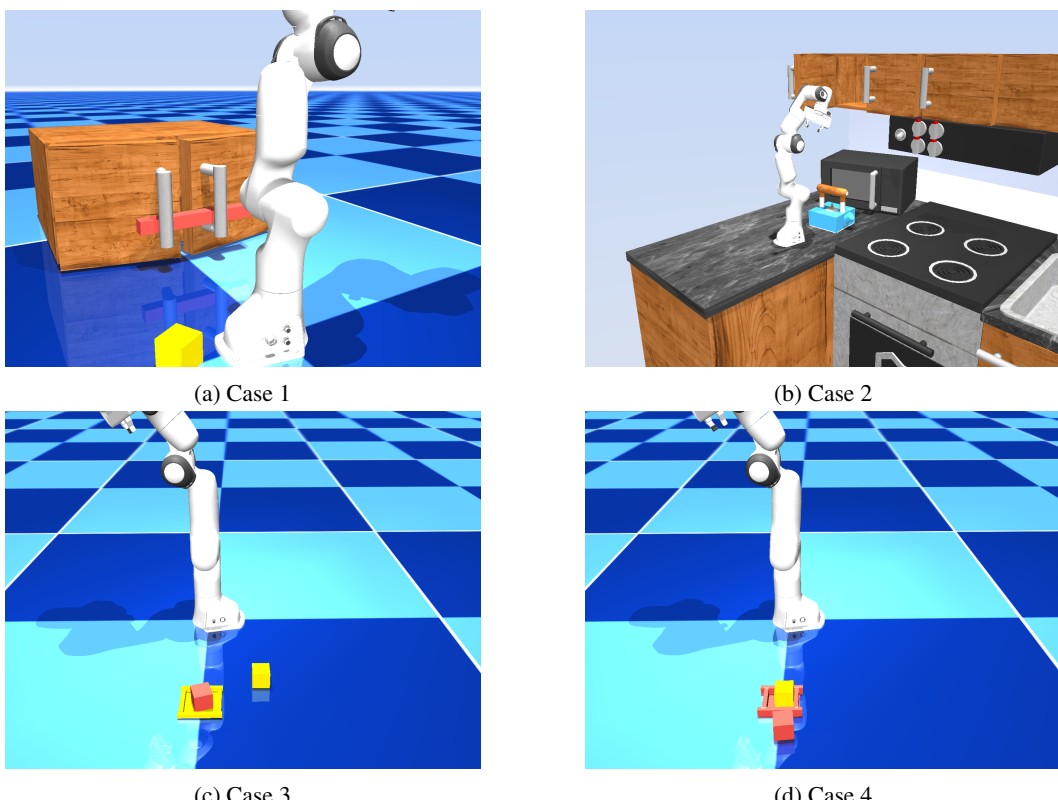

(a) Case 1

(b) Case 2

(c) Case 3

(d) Case 4

Figure C.2: Error case images.

```
reset_reward()
minimize_l2_distance_reward("palm", "red_block_right_side", primary_reward=True)
maximize_l2_distance_reward("red_block_right_side", "target_position_in_cabinet")
execute_plan()
```

The correct primary function should be the second one. As a result, MPC ended prematurely before the robot could remove the bar. The robot was not able to remove the bar in the following steps.

**Case 2 (from Task 4, Figure C.2b)**   The robot tried to open the microwave door but failed due to a kettle obstructing the path. The Perceiver gave five diagnoses, of which three claimed that the kettle was blocking the way, one claimed the cabinet door was blocking the way, and one did not give any conclusive diagnosis. The summary LLM concluded that it was the cabinet door that blocked the action. The robot went on to interact with the cabinet and never removed the kettle.

**Case 3 (from Task 6, Figure C.2c)**   The high-level planner proposed a plan where the first step was *"Identify the cube with the same colour as the crate"*. This was a VLM action. However, the LLM proposed to ask the VLM *"Which cube has the same colour?"*, which was a bit vague, resulting in the VLM answering *"The same color cube is the yellow cube and the yellow cube in the middle of the blue cube group."*. This answer did not provide the necessary information to solve the task. Eventually, the robot put the wrong cube on the crate.

**Case 4 (from Task 7)**   After the robot was not able to execute the task "Place the red cube on the crate", the Perceiver was called to help identify any issues. The Perceiver's diagnoses all mentioned that the robot was holding the red cube but did not identify the yellow cube as blocking the crate, and so the Planner's summary of the VLMs diagnoses was: "Based on the given information, the most probable reason why the robot cannot place the red_cube on the crate is because it is currently holding the red cube." However, it's also important to note that Task 7 used GPT-4V which severly limits the number of ouput tokens from the model, and so a lot of explanations were cut off (for example: "In the image provided, the robot is holding the red cube, which is currently").

## C.4   LLM DIAGNOSIS WITH GROUND-TRUTH DATA

An alternative way to diagnose errors is to input the simulator ground-truth state of the objects to an LLM and ask the LLM to infer a possible reason. We show a scenario here where the cabinet door is locked by a red bar (from Task 3) and test the capability of LLM in this regard. The prompt we use is:

```
A robot is in a simulation environment where it can interact with any object like in the
 real world. The robot would like to open the wooden cabinet but it cannot. Is there
something in this scene preventing that, other than the robot? Assume the robot can
interact with anything. These are the names of the objects in our scene: wooden cabinet,
 red block, yellow cube.

The data of the objects are:
{
    "joints": {
        "wooden_cabinet": 0.16235716765387215
    },
    "objects": {
        "wooden_cabinet_handle": [0.08292539, 0.50163332, 0.21],
        "target_position_in_wooden_cabinet": [0.2,     0.85,    0.0775],
        "red_block_right_side": [0.2158331,  0.55110136, 0.15143364],
        "yellow_cube": [-0.24545329,  0.01631876,  0.03599184],
        "palm": [0.2231031,  -0.00875834,  0.76900653]
    }
}
```

And the response from GPT-4 is:

```
The data does not indicate any object that would prevent the robot from opening the
cabinet. There's no information about any object blocking the cabinet or its handle. The
 problem could be due to the robot itself as it might not possess the necessary
functionality to interact with the objects as intended. The cabinet's joint also has a
non-zero value, which might mean it's not locked or stuck. Hence, there is likely a
mechanical limitation with the robot or an issue with its programming.
```

This example demonstrates that LLM is not capable of directly inferring spatial relations of objects through ground-truth data.

## D   GPT-4V EXPERIMENTS

We run the initial High-Level Planner prompt (Prompt B.2) using GPT-4V[4] on initial task scenes to investigate the ability of GPT-4V to find the correct solution in a single step.

---

[4]https://openai.com/research/gpt-4v-system-card

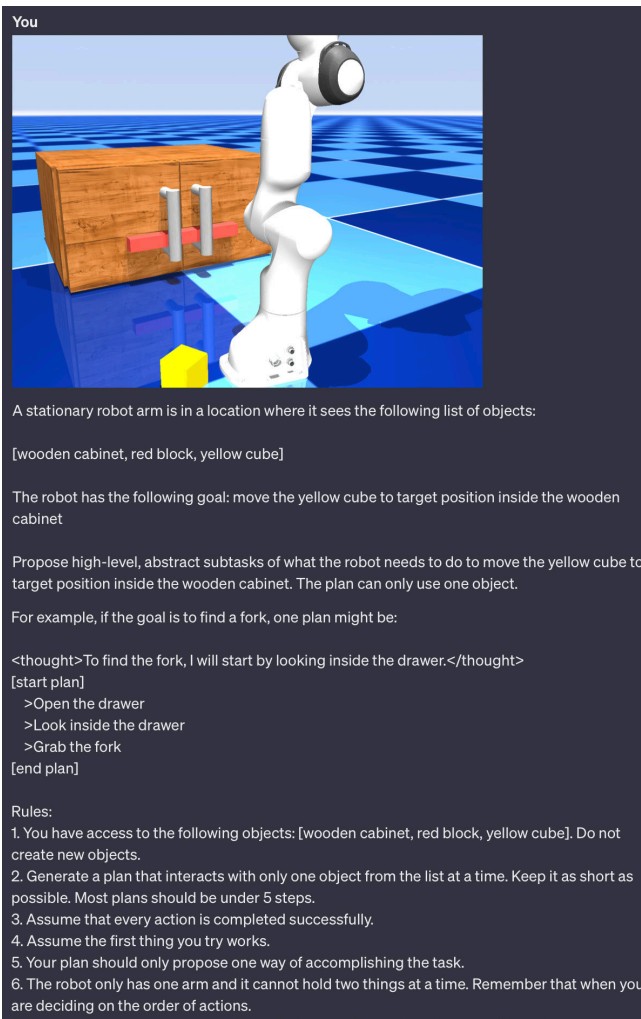

(a) Prompt B.2 to GPT-4V.

**ChatGPT**
<thought>To move the yellow cube to the target position inside the wooden cabinet, I need to consider the simplest and most direct method that involves minimal interaction with other objects. Since the robot has only one arm and cannot hold two things at once, the plan should avoid unnecessary handling of the red block, focusing solely on the yellow cube and the cabinet.</thought>
[start plan]
>Approach the wooden cabinet.
>Open the cabinet door.
>Pick up the yellow cube.
>Place the yellow cube inside the cabinet at the target position.
>Close the cabinet door.
[end plan]

(b) GPT-4V output.

Figure D.1: GPT-4V high-level plan for moving the yellow cube inside the wooden cabinet (Task 3 in Section 4.1.1).

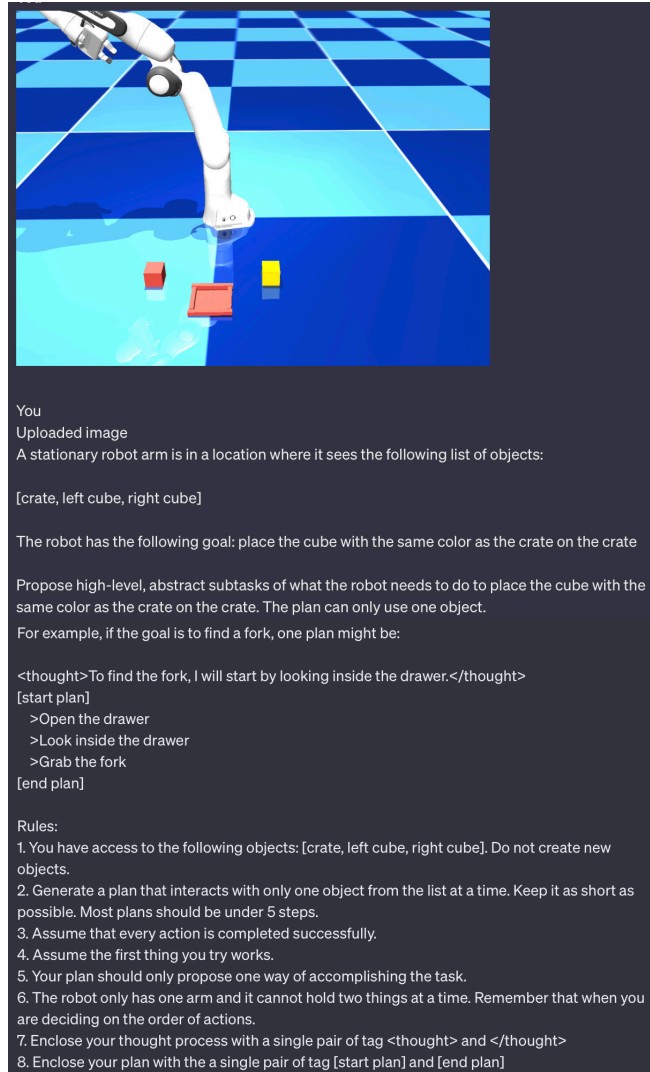

(a) Prompt B.2 to GPT-4V.

(b) GPT-4V output.

Figure D.2: GPT-4V high-level plan for placing the cube with the same color as the crate on the crate (Task 6 in Section 4.1.4).

# E  TAMP EXPERIMENTS

To compare the REPLAN with a TAMP framework, we use Planning Domain Definition Language (PDDL) (McDermott et al., 1998) to define the domain of Cabinet Tasks 1-3 in Table A in Section A as follows:

```
1  (define (domain pick-place-domain)
2    (:requirements :strips :typing :negative-preconditions :conditional-effects)
3
4    ;; Define the object and its possible locations
5    (:types
6        object
7        location
8        conf
9        robot
10       door cabinet cube block - object
11       area
12       remove_area cabinet_area - area
13
14   )
15   ;; define constants
16   (:constants
17       cube_loc cabinet_loc block_loc remove_loc init_loc door_loc open_door_loc -
   location
18       init_conf robot_conf_cube robot_conf_block robot_conf_cabinet robot_conf_remove
   robot_conf_door open_door_conf - conf
19       robot - robot
20       door - door
21       cabinet - cabinet
22       cube - cube
23       block - block
24       remove_area - remove_area
25       cabinet_area - cabinet_area
26   )
27
28   ;; Define predicates
29   (:predicates
30     (at ?obj - object ?loc - location)  ; the Object at location loc
31     (grasped ?obj -object)      ; the object is grasped
32     (at_conf ?conf -conf)              ; the robot is at conf configuration
33     (rob_at_loc ?loc -location)    ; the robot is location loc
34     (is_free ?rob - robot)     ; the robot hand is free
35     (in ?loc -location ?area -area)   ; sampling and certifying the loc location is
   inside an area type
36     (is_closed ?door - door)       ; the door is closed
37     (is_blocked ?door)           ; the door is blocked
38     (is_moveable ?init_conf -conf ?final_loc -location ?final_conf - conf)
39     ; certifying the robot can move from the initial condition to the goal location/pose
    with the sampled final configuration
40
41   )
42
43     ;; Define actions
44   (:action remove
45       :parameters (?block - block ?door - door ?rob - robot ?init_conf ?final_conf -
   conf ?init_loc ?final_loc -location)
46       :precondition (and
47           (is_blocked ?door)
48           (grasped ?block)
49           (not (is_free ?rob))
50           (at_conf ?init_conf )
51           (at ?block ?init_loc)
52           (rob_at_loc ?init_loc)
53           (in ?final_loc remove_area)
54           (is_moveable ?init_conf ?final_loc ?final_conf )
55       )
56       :effect (and
57           (rob_at_loc ?final_loc )
58           (not (rob_at_loc ?init_loc))
59           (at ?block ?final_loc)
60           (at_conf ?final_conf )
61           (is_free ?rob)
62           (not (grasped ?block))
63           (not (is_blocked ?door))
64           (not (at_conf ?init_conf ))
65       )
66   )
67
68       (:action place
```

```
69        :parameters (?cube - cube ?door - door ?rob - robot ?init_conf ?final_conf - conf
   ?init_loc ?final_loc -location)
70        :precondition (and
71            (not (is_closed ?door))
72            (not (is_blocked ?door))
73            (grasped ?cube)
74            (not (is_free ?rob))
75            (at_conf ?init_conf )
76            (at ?cube ?init_loc)
77            (rob_at_loc ?init_loc)
78            (in ?final_loc cabinet_area)
79            (is_moveable ?init_conf ?final_loc ?final_conf )
80        )
81        :effect (and
82            (rob_at_loc ?final_loc )
83            (at ?cube ?final_loc)
84            (at_conf ?final_conf )
85            (not (at_conf ?init_conf ))
86            (is_free ?rob)
87            (not (grasped ?cube))
88        )
89    )
90

91
92    (:action pick
93      :parameters (?init_conf ?final_conf -conf ?obj - object ?loc - location ?rob -robot)
94      :precondition (and
95        (at_conf ?init_conf )
96        (at ?obj ?loc)
97        (is_free ?rob )
98        (is_moveable ?init_conf ?loc ?final_conf )
99        )
100     :effect (and
101       (at_conf ?final_conf)
102       (not (at_conf ?init_conf))
103       (rob_at_loc ?loc)
104       (not (is_free ?rob))
105       (grasped ?obj)
106     )
107   )

108
109   (:action open
110       :parameters (?init_conf -conf ?door -door ?rob -robot )
111       :precondition (and
112         (is_closed ?door)
113         (not (is_blocked ?door))
114         (grasped ?door)
115         (not (is_free ?rob))
116         (at_conf ?init_conf )
117         (is_moveable ?init_conf open_door_loc open_door_conf )
118       )
119       :effect (and
120         (not (is_closed ?door))
121         (not (grasped ?door))
122         (is_free ?rob)
123         (not (at_conf ?init_conf ))
124         (at_conf open_door_conf)
125         (rob_at_loc open_door_loc)
126         (at door open_door_loc)
127         )
128     )

129
130     (:action is_moveable_cube
131         :parameters ()
132         :precondition (and
133           (not (is_closed door))
134           (not (is_blocked door))
135         )
136         :effect (is_moveable robot_conf_cube cabinet_loc robot_conf_cabinet)
137     )

138
139 )
```

Snippet 1: PDDL domain definition for Tasks 1-3.

Tasks 1-3 problem definition are written as follows:

```
1
2 (define (problem pick-place-problem)
3   (:domain pick-place-domain)
```

```
4
5    ;; Define objects
6    (:objects
7
8    )
9
10   ;; Define initial state
11   (:init
12     (at cube cube_loc)
13     (at door door_loc)
14     (at block block_loc)
15     (at_conf init_conf)
16     (rob_at_loc init_loc)
17     (in remove_loc remove_area)
18     (in cabinet_loc cabinet_area)
19     (is_free robot)
20     (is_moveable init_conf cube_loc robot_conf_cube)
21     (is_moveable open_door_conf cube_loc robot_conf_cube)
22     (is_moveable robot_conf_block remove_loc robot_conf_remove)
23     (is_moveable robot_conf_remove door_loc robot_conf_door)
24     (is_moveable init_conf block_loc robot_conf_block)
25     (is_moveable init_conf door_loc robot_conf_door)
26     (is_moveable robot_conf_door open_door_loc open_door_conf)
27
28     ;=======================================================================
29     ;; commenting the following two initial conditions can change the robot behavior
   greatly in terms of task plan
30     ;; easy mode (Task 1): comment both of the following lines [comment (is_closed door)
    and (is_blocked door)]
31     ;; hard mode (Task 2): comment the second condition [(is_blocked door)]
32     ;; expert mode (Task 3): keep both of the following conditions uncommented.
33     ;=======================================================================
34
35     ; door is closed at the begining
36     (is_closed door)
37     ; door is blocked at the begining
38     (is_blocked door)
39
40
41   )
42
43   ;; Define goal
44   (:goal
45     (and ;; only picking the cube
46         ; (grasped block)
47
48         ;; picking and placing the cube inside the cabinet
49         (at cube cabinet_loc)
50           ; (at block remove_loc)
51           ; (grasped door)
52
53
54
55     )
56   )
57 )
```

Snippet 2: PDDL problem definition for Cabinet Tasks 1-3.

The cabinet task problem is solved using the lpg-td (Gerevini et al., 2006) solver from the planutils library[5].

As seen in Snippets 1 and 2, even for three cabinet tasks 1-3, domain and problem definitions require careful and laborious attention. Task-solving details are outlined in Snippets with blue and olive colours. While our method discovers information through interaction and reasoning over perceiver's feedback, the PDDL solver relies on ground truth (highlighted in blue) and rules (example in olive) provided by the user for problem resolution.

For the same task, PDDLStream (Garrett et al., 2020) offers an alternative using a Task and Motion Planning (TAMP) framework. Rather than human-grounded truth information, a motion planning framework certifies predicates through *streams*. However, this requires user-defined rules for success or failure and a motion planner. Solving long-horizon problems with PDDLStream may become computationally expensive (Khodeir et al., 2023).

---

[5]https://github.com/AI-Planning/planutils

REPLAN can robustly solve long-horizon multi-stage problems through interaction with the environment and reasoning based on the perceiver's feedback. This capability enables REPLAN to uncover underlying rules without the need for an additional domain description and ground truth information.

