# OpenReview forum: "RePLan: Robotic Replanning with Perception and Language Models"
_ICLR.cc/2024/Conference — Submitted to ICLR 2024_

### Official Review · Reviewer_6UHr · 2023-10-23

**Soundness:** 2 fair
**Presentation:** 2 fair
**Contribution:** 2 fair
**Rating:** 5
**Confidence:** 4

**Summary:**

The authors build off recent work on controlling robots via large language models and vision language models. The closest analogue of their work is recent lines of work on generating reward functions via language models (i.e. Zero-Shot Reward Specification via Grounded Natural Language, Language to Rewards for Robotics Skill Synthesis, etc.)

In general, prior work uses a prompted LLM to generate a reward function based on a natural language command from the user for what task they want to perform, and this reward is optimized by MPC to generate low level actions.

This work augments this flow by including a VLM "Perceiver". After the robot acts according to the generated reward, this perceiver is given an image of the scene + an instruction to either confirm the task is completed or generate a text description of why it has failed. If the robot has failed, this text description feedback is fed back into the high-level planner to generate a new instruction + new reward function. This is the "replanning" step of their work.

**Strengths:**

The core idea of using a VLM to generate textual feedback for a model makes sense and is a good addition to this research direction in robot learning. The videos are very helpful for showcasing the resulting method.

**Weaknesses:**

Although the high level idea of the paper is reasonable, it is both not evaluated as extensively as would be helpful and is not put in context of recent literature very well.

As a minor note: please fix the references to surround the authors of citations in parentheses. This would make the paper much easier to read.

On the experimental front:

* the exact prompts used do not appear in the paper or appendix, which makes it hard to judge how the LLM was told to incorporate past feedback
* The evaluation setting is 4 tasks, done for 3 trials each. This is an incredibly small number of tasks and trials for the method. As a point of comparison: the Language to Rewards paper used as a baseline was tested on 17 tasks with 10 generated rewards functions per task, run through MPC 50 times each. My understanding is that in Table 1, RePLan without replanning is almost identical to Language to Rewards, but is only successful 1/3 times in the easiest Task 1 setting, compared to 3/3 from Language to Rewards. To me this seems like it is just because of random noise, but if that's the case, why should we trust any of the other numbers in the table? This is the point of issue I find most important about the paper.

On the prior literature front:

There are number of prior works based on providing LLM feedback from the environment, either ground truth or from VLMs. Examples are Voyager by Wang et al, Inner Monologue by Huang et al, and Towards a Unified Agent with Foundation Models by Di Palo et al. I would appreciate some discussion about such lines of work, since to me it is not so clear if this is doing anything very different from these works. I believe at most you can argue that this paper is using MPC instead of RL or imitation learning, but otherwise prior work has used chain-of-thought style prompting to decompose high-level language to low level language, generate rewards from said low level language, provide feedback via VLMs, etc. That is not to say that the combination of prior work cannot be novel, but in this instance, it does not feel like much is coming from said combination. Especially given the weakness of the experimental results.

Edit: some more experimental results were provided and I have adjusted score from 3 -> 5.

**Questions:**

When doing replanning, is there any cap on how many iterations of replanning the agent is allowed to do? Could the authors also discuss if they see failure cases from the Perceiver, or in deciding if a task is completed or not?

---

> ### Author Response · Authors · 2023-11-18
>
> Thank you very much for your feedback! To address your concerns:
>
> **1. Missing prompts.** We apologize for not originally including the prompts used. In our updated draft, we have included all prompts used in Appendix B and indicate how they are used in Algorithm 1 and throughout the text.
>
> **2. Limited evaluations.** We agree with the reviewer that our experiment setting was limited. We have increased the number of experiments on two fronts. The first is that we added two new environments and three new tasks. The first added environment contains a cabinet that is initially locked, and a lever. The robot has to open the cabinet to deposit an item in there, but it cannot. There is also nothing physically blocking it, and so the robot has to reason that it must somehow interact with the lever in order to unlock the cabinet. The second environment is a scene with a red crate and two cubes, one red and one yellow. In the first task, the robot must place the cube with the same colour as the crate on the crate. This requires conditional reasoning with the Perceiver. The second tasks also requires the robot to place the red cube on the crate, but this time there is a physical blocker: the yellow cube is already on the crate, and there is only room for one. Thus, the robot must first remove the yellow cube in order to accomplish the task. We have also increased the number of runs of each experiment from 3 to 10. We also include a breakdown of the number of actions the robot does in order to do the tasks. In total, the robot performs 401 MPC actions and 101 VLM actions. This is more than previous work (L2R, 170 MPC actions). To address the reviewer’s concerns about experimental errors, we include a more thorough breakdown of model components and an error analysis (see Appendix C2 and C3).
>
> **3. Prior work.** Thank you to the reviewer for pointing out these prior works. Indeed, we include many of them in our updated draft to clarify our contributions. Specifically, Inner Monologue (IM) does not actively use feedback from VLMs to replan during a task; it uses feedback from humans or object recognition models to know what objects are present or absent (this would be most similar to row 3 in our Table 1). IM uses a success detector in order to determine whether it could resample the low-level policy for a pick-and-place action, but it would not be able to handle obstacles and does not reason on why the action failed. It has been shown in works since binary success detection is not enough to perform complex tasks [1]. Voyager also shows this by receiving feedback from a chat bot in the Minecraft API on why code execution errors were generated. They (in addition to other works) also show that Verifiers are important for long-horizon task completion. We agree that the use of Verifiers and environment feedback is extremely important. We have added these references to our Related Works section. Towards a Unified Agent relies on previously collected observations to finetune a CLIP reward model. Our method does not require any model training. We believe our method is novel because we combine several key insights in order to execute complicated, long-horizon tasks without the need for model re-training or human guidance. First, we use perceiver models to diagnose problems and gather information about object states in order to replan if the robot encounters issues. We create hierarchical plans for both high-level reasoning and low-level reward generation, allowing for adaptive robot behaviour that has only been previously shown on single-step tasks. Finally, we use verifiers at both high-level and low-level reasoning to enable long-horizon tasks of up to 17 steps.
>
> **4. Replanning iterations.** Our replanning framework is set up recursively, so that the agent can replan when performing at any subtask. However, we set a threshold on recursion depth (r=2) and the number of replanning stages allowed (p=2) to lower runtime and API call costs.
>
> **5. Failure cases.** We include a more in-depth error analysis of all components in our system in this draft. Specifically, we showcase Perceiver errors in Appendix C3. The Perceiver is often able to successfully identify what the reason for plan failure is, but sometimes the interaction between the Perceiver and Planner results in the wrong summary reason. We also include an ablation study in C2 that reports the object recognition and object reasoning performance of two state-of-the-art open-source VLMs and GPT-4V.
>
> **6. Citations formatting.** We apologize for the citation issue and have fixed it in the current draft, we thank the reviewer for pointing it out to us.
>
>
> [1] Lei Wang et al, “A Survey on Language Language Model based Autonomous Agents”. (2023)

---

> > ### Comment · Reviewer_6UHr · 2023-11-22
> >
> > Thanks for replying to concerns. I will increase the score of my review. I do agree with reviewer b9qX that the Perceiver should be analyzed in more detail as it is the more important part of the work.

---

> > > ### Author Response · Authors · 2023-11-22
> > > **VLM Perceiver analysis**
> > >
> > > Dear Reviewer 6UHr,
> > >
> > > Thank you for checking our responses and thank you for increasing your score.  We agree with you and Reviewer b9qX that the Perceiver analysis was insufficient in the previous version of the paper. We have added a detailed analysis of the Perceiver in Appendix C. We have also added extensive explanation of the VLM Perceiver in the paper and how RePlan leverages the VLM for long-horizon planning.
> > >
> > > Specifically, we show a **quantitative analysis** that measures the performance of **three state-of-the-art VLMs (Qwen, Llava, GPT-4V)** in two tasks: **object recognition and object reasoning**. We provide the results below and also in Append C2. We also show a **qualitative anlysis** below and in Appendix C3 that demonstrates sample VLM inputs/ouputs and errors. Finally, we show a **motivating example** of why constrained VLM prompting is needed since GPT-4V is not able to solve the tasks out-of-the-box.
> > >
> > > VLMs are known to be good for object detection, however their reasoning capabilities are not clear. In order to avoid relying on a single VLM prompt, we use several prompts and ask an LLM to create a consensus from their outputs based on the LLM's knowledge of the scene and robot goals. This also allowed the LLM to consider the consistency among the VLM replies (e.g. if 3/6 VLM outputs identify the same problem in the scene, it is more likely that those outputs are reflective of the scene). The motivation behind this was that VLMs are good at captioning images in VQA settings -- in our setting, we ask it to "caption" a problem. This combination allowed for effective reasoning capabilities.
> > >
> > > **QUANTITATIVE ANALYSIS:**
> > >
> > > We do this study because our Perceiver has two fundamental roles: object detection and failure diagnosis.  Object detection refers to the percentage of objects the VLM is able to recognize in the scene. For object reasoning, we gave the VLM 6 prompts related to failure detection (which we show below and in Appendix B, Figure 9). During the RePLan pipeline, we prompt the VLM with all 6 prompts and then ask the LLM High-Level Planner (which has better reasoning skills) to, given the 6 answers and the LLM's knowledge of the scene (e.g. what objects are present, that a robot is doing a task in a simulation) to provide the most likely reason for why the robot can’t do the task (this prompt is shown below and in Appendix B, Figure 12). First, we report the percentage of responses from the VLM that encompass the true reason out of the 6 inputs for the robot not being able to do the action. Then, we report the ability of the LLM to summarize the VLM outputs into a correct/likely reason as to why the robot cannot perform the task. We repeated each ablation 5 times.
> > >
> > > ## Task 3
> > >
> > > | Scenarios                                 | Qwen + SAM | Qwen  | Llava | GPT-4V |
> > > |-------------------------------------------|------------|-------|-------|--------|
> > > | VLM object detection                      | $100\%$    | $66\%$| $100\%$ | $100\%$ |
> > > | VLM Reasoning                             | $67\%$     | $0\%$ | $23\%$ | $100\%$ |
> > > | LLM summarization and consistency step    | $100\%$    | $0\%$ | $100\%$ | $100\%$ |
> > >
> > >
> > > ## Task 4
> > >
> > > | Scenarios                                 | Qwen + SAM | Qwen  | Llava | GPT-4V |
> > > |-------------------------------------------|------------|-------|-------|--------|
> > > | VLM object detection                      | $100\%$    | $100\%$| $100\%$ | $100\%$ |
> > > | VLM Reasoning                             | $50\%$     | $66\%$| $40\%$ | $83\%$ |
> > > | LLM summarization and consistency step    | $100\%$    | $100\%$| $20\%$ | $100\%$ |
> > >
> > >
> > > We found that Qwen somtimes struggled with object detection of smaller objects, and so we coupled it with Segment Anything Model (SAM) [1] to first segment the objects in the scene. We found that all models did well with object recognition (except for Qwen when not used with SAM). For object reasoning, Qwen + SAM was able to get the correct scene error in 50-67\% of the prompts it was given, and then the LLM was able to summarize the prompts to generate the correct error reason overall. The reason the LLM was able to do this despite the VLM not giving perfect answers was that the remaining VLM answers pertained where the robot was located or a general comment about the objects in the scene. Llava tended to reply that it was unable to reason because the scene was a simulation and not real life. GPT-4V had the best overall performance in all categories, but API calls to it are still restricted.
> > >
> > > (continued in next response)

---

> > > > ### Author Response · Authors · 2023-11-22
> > > > **VLM Perceiver analysis, part 2**
> > > >
> > > > **QUALITATIVE ANALYSIS:**
> > > >
> > > > We also show some examples below of prompts to the VLM and its responses, as well as the LLM summary. We will include them in the draft before the deadline, but are posting them here now:
> > > >
> > > > **Task 3: VLM is asked to diagnose why the robot cannot open the cabinet.**
> > > >
> > > > Input to VLM:
> > > > ```
> > > > 1. A robot is in a simulation environment where it can interact with any object like in the real world. The robot would like to open the wooden cabinet but it cannot. Is there something in this scene preventing that, other than the robot? Assume the robot can interact with anything. These are the names of the objects in our scene: wooden cabinet, red block, yellow cube
> > > > 2. In a simulation, a robot wants to open the wooden cabinet but can't. Is anything else, besides the robot, blocking it? Check the objects in the scene: wooden cabinet, red block, yellow cube.
> > > > 3. Robot in a simulation wants to open the wooden cabinet, can't. Something else stopping it? Objects in scene: wooden cabinet, red block, yellow cube.
> > > > 4. A robot can engage with any item. It wants to open the wooden cabinet but can't. Is an object in this scene, apart from the robot, hindering it? Objects present: wooden cabinet, red block, yellow cube
> > > > 5. I would like to open the wooden cabinet but I cannot. Is there something in this scene preventing that, other than the robot? These are the objects in the scene: wooden cabinet, red block, yellow cube
> > > > 6. I would like to open the wooden cabinet but I am unable to. Is there something in this scene preventing me from doing that? Ignore the robot. These are the names of the objects: wooden cabinet, red block, yellow cube
> > > > ```
> > > > Outputs from VLM:
> > > > ```
> > > > 1. There is a red block in front of the wooden cabinet, preventing the robot from opening it.
> > > > 2. Yes, the red block and the yellow cube are blocking the robot from opening the wooden cabinet.
> > > > 3. The robot in the simulation is unable to open the wooden cabinet because it is facing the wrong direction. It is facing the cabinet, but the door is closed. The robot is also unable to see the red block and the yellow cube that are located outside of the cabinet.
> > > > 4. There is no existence of any other object in the scene, apart from the robot, the wooden cabinet, the red block, and the yellow cube.
> > > > 5. There is a robot standing in front of the wooden cabinet, preventing it from being opened.
> > > > 6. There is a robot standing in front of the wooden cabinet, preventing you from opening it.
> > > > ```
> > > > LLM Summary:
> > > > ```
> > > > The most probable reason why the robot cannot open the wooden cabinet is that there is a red block in front of the cabinet, blocking its path.
> > > > ```
> > > >
> > > > (continued in next response)

---

> ### Author Response · Authors · 2023-11-22
> **VLM Perceiver analysis, part 3**
>
> **Task 4: VLM is asked to diagnose why the robot cannot open the microwave.**
>
>
> Input to VLM:
>
> ```
> 1. A robot is in a simulation environment where it can interact with any object like in the real world. The robot would like to open the microwave but it cannot. Is there something in this scene preventing that, other than the robot? Assume the robot can interact with anything. These are the names of the objects in our scene: blue kettle, microwave, cabinet
> 2. In a simulation, a robot wants to open the microwave but can't. Is anything else, besides the robot, blocking it? Check the objects in the scene: blue kettle, microwave, cabinet.
> 3. Robot in a simulation wants to open the microwave, can't. Something else stopping it? Objects in scene: blue kettle, microwave, cabinet.
> 4. A robot can engage with any item. It wants to open the microwave but can't. Is an object in this scene, apart from the robot, hindering it? Objects present: blue kettle, microwave, cabinet
> 5. I would like to open the microwave but I cannot. Is there something in this scene preventing that, other than the robot? These are the objects in the scene: blue kettle, microwave, cabinet
> 6. I would like to open the microwave but I am unable to. Is there something in this scene preventing me from doing that? Ignore the robot. These are the names of the objects: blue kettle, microwave, cabinet
> ```
>
> Outputs from VLM:
>
> ```
> 1. There is a blue kettle in the scene, which is preventing the robot from opening the microwave.
> 2. In the scene, there is a blue_kettle on the microwave and a cabinet above the microwave. The robot is not blocking the microwave, so the only thing preventing the robot from opening the microwave is the blue_kettle on the microwave.
> 3. The robot in the scene is unable to open the cabinet because there is a blue_kettle placed on the surface in front of the cabinet. This object is blocking the robot's access to the cabinet, preventing it from opening it.
> 4. There is no existence of a blue kettle in the image description.
> 5. There is a blue kettle on the countertop that could be preventing you from opening the microwave.
> 6. There is a robot standing on the counter in front of the microwave. It is blocking access to the microwave.
> ```
>
> LLM Summary:
> ```
> The most probable reason why the robot cannot open the microwave is that there is a blue kettle placed on the microwave, blocking the robot's access to it.
> ```
>
> **In Appendix C3, we also show some error cases of where the interaction betewen the Perceiver and High-Level Planner fails.** As an example, in Task 4, the robot tried to open the microwave door but failed due to a kettle obstructing the path. The Perceiver gave five diagnoses, of which three claimed that the kettle was blocking the way, one claimed the cabinet door was blocking the way, and one did not give any conclusive diagnosis. The summary LLM concluded that it was the cabinet door that blocked the action. The robot went on to interact with the cabinet and never removed the kettle. However, in 9 runs out of 10, the High-LeveL Planner was able to correctly summarize the reason from the VLM outputs.
>
> **PERCEIVER-PLANNER MOTIVATION**
>
> As an additional experiment, we also report the ability of GPT-4V to solve Tasks 3 and 6 out-of-the-box by prompting it to create a plan for opening the cabinet and placing the cube with the same colour as the crate on the crate (Appendix D). In both cases, the generated plans did not provide a working solution. In the first case, plan did not identify that there was a block across the handles of the cabinet the robot needed to first remove, and in the second case, the plan did not specify the name of the cube to use. This shows combining the VLM with specific LLM prompting is essential in order to solve these tasks.
>
> We again thank you for your feedback, please let us know if there are additional analyses you wish to see.
>
> [1] Kirillov A et al, Segment Anything. (2023)

---

### Official Review · Reviewer_Ya9u · 2023-11-01

**Soundness:** 2 fair
**Presentation:** 3 good
**Contribution:** 3 good
**Rating:** 3
**Confidence:** 4

**Summary:**

The paper presents a hierarchical method for solving multi-stage manipulation tasks with language models, consisting of a LLM-based high-level planner as well as a LLM-based low-level reward/cost generator. The high-level planner decomposes the task into multiple stages in natural language, with the option to replan based on perceptual input if needed. The low-level reward generator takes as input the decomposed sub-task and generates reward for an MPC controller, which uses predictive sampling in MuJoCo to generate low-level robot actions. The method is evaluated on 4 tasks in 2 scenes, where it is compared against a recent baseline “Language to Reward” and ablated across the method’s different components.

**Strengths:**

- The presented idea is clear and well-motivated — leveraging LLMs in a hierarchical framework for both high-level task planning and low-level motion planning. Compared to prior work, “Language to Reward”, it is clear that such hierarchical approach is needed for long-horizon tasks and can also offer additional robustness as the system can replan its high-level action.
- The literature review is also thorough, covering many recent works in this domain. However, this part can be improved because it is now more like a laundry list instead of putting the work in the context of prior works.

**Weaknesses:**

- Currently the biggest limitation seems to be the lack of thorough experiments, which can use some improvement along two axes. One is the breadth of the tasks: there are only four tasks investigated in this work while there are also quite some similarities between them. An important advantage of using LLMs is that it is possible to apply to a wider set of tasks more easily. The other axis is the quantitative evaluation: currrently only 3 runs are performed for each entry in Table 1, which makes the quantitative results not very convincing as it is also pointed out in the paper that there is “high variance of completion”. In addition, the paper does not compare to prior methods that are not based on LLMs, e.g., task and motion planning methods or hierarchical RL methods.
- Another limitation lies in the use of the simulator ground-truth for MPC. This raises the question whether the approach can be applicable to real-world settings. However, the high-level planning part does use VLM for grounding image observations, but it’s unclear how this can be achieved for the attributes referred in MPC, e.g. “block_r_side”, “cabinet_handle”.
- Currently the intro reads more like related works, where it may be confusing to readers what the actual motivation of the work is. More care can be taken to improve the intro while appropriately contextualizing the work.
- The citation format in many places are currently incorrect — parenthesis often should be used.

**Questions:**

What is the prompt being used for LLMs? Can the authors provide more examples of paired LLM output and environment execution?

---

> ### Author Response · Authors · 2023-11-18
>
> We are glad that the reviewer finds our idea clear and well-motivated. We also appreciate the reviewer’s insightful feedback. Below we address the reviewer’s concerns:
>
> **1. Improved literature review.** We agree with the reviewer’s observation that the literature review could be improved. We have updated our introduction section and related works section to provide a more thorough and structured literature review, and have re-written sections to better put our work in the context of previous papers and ideas.
>
> **2. Lack of thorough experiments and comparison with prior non-LLM works.** We appreciate the reviewer for their suggestion to improve our evaluations. We have worked on both of the axes by including three new tasks in two new scenes and increasing the number of runs from three to ten. One of the new tasks involves reasoning about an interaction with a lever-locking mechanism, and the other new tasks emphasize perceiving and reasoning about colors and executing conditional plans. We believe the inclusion of these new tasks can improve the variety of our evaluation. We also improve the analysis of existing tasks in the Results section and  Appendix C by analyzing the number of actions required by our tasks (401 MPC actions + 101 VLM actions in total), as well as analyzing common errors. Finally, we include two additional baselines for a few tasks: PDDL (Appendix E) and GPT-4V (Appendix D). We find that out-of-the-box, GPT-4V is not able to solve our tasks. We include a PDDL domain and problem definition for Tasks 1-3. We show that they are not trivial to write and rely on human-provided ground truth information (for example, that the door is blocked). We also consider using PDDLStream to solve the problem. In this case, some of the human-provided ground-truth information is certified by Stream (by using a motion planner it verifies the feasibility of action execution at the planning time); however, it’s inefficient to solve. In terms of hierarchical RL, our work assumes text-based goals with no access to pretrained skills. To the best of our knowledge, there are no prior RL works that succeed under these assumptions. The closest one we find is [1], which is only evaluated in single-motion tasks, while the majority of our tasks are multi-step. In addition, our algorithm can be evaluated on the fly while RL methods typically require extensive training, which we believe can be an unfair comparison.
>
> **3. Simulator ground truth for MPC.** We believe using simulator ground truth is a necessary compromise to make, and is currently what state-of-the-art works use (Language to Rewards [2], Inner Monologue [3]). Using a ground truth simulator does not jeopardize transferring the system to the real world. For example, [2] and [3] utilize a vision model to map the real-world objects to the simulator and then execute the simulator-produced actions in real world. Another possible way could be collect demonstrations in the simulator and distill an image-based policy from them, which can be directly applied to the real world.
>
> **4. The introduction reads more like related works.** We apologize for the possible confusion. We have updated our introduction to provide a better motivation for our work. We have also fixed the citation formatting issue. We thank the reviewer for pointing this out.
>
> **5. Missing prompts.** We have included the detailed prompts in Appendix B in our updated manuscript, as well as how they are used in Algorithm 1 and throughout the text. We hope it can provide enough information for the reviewer.
>
> - [1] Wenhao Yu et al, “Language to Rewards for Robotic Skill Synthesis”. (2023)
> - [2] Tianbao Xie et al, “Text2Reward: Automated Dense Reward Function Generation for Reinforcement Learning”. (2023)
> - [3] Wenlong Huang et al, “Inner Monologue: Embodied Reasoning through Planning with Language Models”. (2023)

---

> > ### Comment · Reviewer_Ya9u · 2023-11-23
> > **Thanks for the response**
> >
> > I would like to thank the authors for their response. I believe the updated draft has addressed most of my previous concerns except for "using simulator ground-truth in MPC":
> >
> > - The reason I'm raising this concern is because an important claimed contribution of this work is using a VLM to perform *image-based* reasoning for long-horizon tasks. However, the low-level actions still rely on a ground-truth model of the environment, which makes the assumption unclear and defeats the purpose of using VLMs in the first place. If a ground-truth model is assumed to be available (e.g., the joint angle for drawers is always known), one can easily just use this to feed into a text-only LLM to reason about replanning actions, which is likely more accurate than using VLMs anyway.
> >
> > - While I agree with authors' response that prior works (e.g., [1]) also assume the ground-truth model is available, these works *do* show the feasibility of transferring to the real world (or even just image-based observations in sim) on certain tasks. But currently this is not shown in the paper.
> >
> > - Furthermore, assuming ground-truth models is also acceptable if this is not inherently tied to the contributions of the paper. For example, the claim of the paper can be rephrased to only focusing on high-level replanning using VLMs (this should also be well contextualized in the current literature). However, the current claimed contribution is that the framework also enables replanning at the low-level actions, which largely rely on prior work ([1]) and does not offer additional insights to the community.
> >
> > Due to the above reasons, I'm only able to raise my recommendation to 4.

---

> > > ### Author Response · Authors · 2023-11-23
> > >
> > > Dear Reviewer Ya9u,
> > >
> > > Thank you for checking our responses and for increasing your score.
> > >
> > > We kindly wish to bring to the reviewer's attention that our contribution is a framework that can handle long-term, complex multi-stage tasks, which consists of four essential components:
> > >
> > > - using perceiver models for high-level replanning,
> > > - creating hierarchical plans with language models,
> > > - verifying outputs from these language models,
> > > - robot behavior through reward generation.
> > >
> > > While high-level replanning is one of the major compoments of our proposed method, RePLan is an **end-to-end framework** that solves long-horizon planning tasks and outputs low-level robot controls (7-dof joint and gripper values in the case of Franka arms)
> > >
> > >
> > > - Regarding the first point about ground-truth data, the reviewer stated that "one can easily just use this (ground-truth data) to feed into a text-only LLM to reason about replanning actions". We observe, after doing experiments, that this is not the case. Below, we show an example below that using an LLM and ground-truth data only is not sufficient to complete our replanning task, as opposed to using VLM with scene images (which we also added to Appendix C.4). Even though in theory ground-truth data should provide enough information to understand the scene by the Planner, as far as we understand, this has not been backed up by sufficient evidence and we have not found it to be the case either. Integrating the Perceiver allows the Planner to access visual information that it cannot easily determine from positions and angles alone. If the reviewer has some evidence supporting that ground-truth data is sufficient to "easily" perform high-level replanning and multi-step reasoning, we are happy to perform a comparision between our method and the ones using only ground-truth data. The example is as follows:
> > >
> > > ```
> > > A robot is in a simulation environment where it can interact with any object like in the real world.
> > > The robot would like to open the wooden cabinet but it cannot.
> > > Is there something in this scene preventing that, other than the robot? Assume the robot can interact with anything.
> > > These are the names of the objects in our scene: wooden cabinet, red block, yellow cube.
> > > The data of the objects are:
> > > {
> > >     "joints": {
> > >         "wooden_cabinet": 0.16235716765387215
> > >     },
> > >     "objects": {
> > >         "wooden_cabinet_handle": [0.08292539, 0.50163332, 0.21],
> > >         "target_position_in_wooden_cabinet": [0.2,    0.85,   0.0775],
> > >         "red_block_right_side": [0.2158331,  0.55110136, 0.15143364],
> > >         "yellow_cube": [-0.24545329,  0.01631876,  0.03599184],
> > >         "palm": [0.2231031m  -0.00875834m  0.76900653]
> > >     }
> > > }
> > > ```
> > >
> > > ```
> > > The data does not indicate any object that would prevent the robot from opening the cabinet.
> > > There's no information about any object blocking the cabinet or its handle.
> > > The problem could be due to the robot itself as it might not possess the necessary functionality to interact with the objects as intended.
> > > The cabinet's joint also has a non-zero value, which might mean it's not locked or stuck.
> > > Hence, there is likely a mechanical limitation with the robot or an issue with its programming.
> > > ```
> > >
> > > (continued in next post)

---

> > > > ### Author Response · Authors · 2023-11-23
> > > >
> > > > - Regarding the second point concerning the sim2real gap and the application of our method in real-world scenarios, it's important to clarify that we do not rely on ground-truth data for high-level replanning, a key aspect of our contribution. While we do employ ground-truth data for low-level robot motion planning, we acknowledge the limitation that this may impose on the applicability of RePlan in real-world experiments. It's worth noting that many papers, as the reviewer is aware, have focused on object pose estimation for simple objects in the context of low-level robot motion planning. These existing approaches can be integrated into our system. However, for more intricate scenarios, such as grasping a blocker on a cabinet door, advanced techniques are necessary to comprehend object affordances for manipulation. For instance, in the work by L. Wang et al., titled "Self-Supervised Learning of Action Affordances as Interaction Modes" (ICRA 2023), the authors delve into such complexities. The examples cited in [1] (Language-to-Reward) involve tasks like picking a cube/apple and opening a drawer. We posit that addressing the challenge of utilizing visual data for robot motion planning in intricate, contact-rich manipulation scenarios, as in our case, demands dedicated research efforts.
> > > >
> > > >
> > > > - Regarding the third point, we apologize if our contribution description was unclear. We have revised it to explicitly state that replanning occurs exclusively at the high-level task planning. It's important to clarify that while our method is end-to-end, covering high-level task description to low-level motion generation, there is no explicit replanning at the low-level. Upon thorough review, we confirmed that our manuscript does not make any claims about replanning at the low-level motion of the robot. Our approach for low-level reward generation for Model Predictive Control (MPC) draws inspiration from the work in [1] (Language to Rewards). Upon careful examination of [1], it appears they do not engage in replanning at the motion level. Instead, their motion policy generator operates in a closed-loop (reactive) manner, aligning with our approach.
> > > >
> > > >
> > > > Thank you once more for your feedback. Please feel free to ask any additional questions for further clarification.
> > > >
> > > > Best,
> > > > Authors

---

### Official Review · Reviewer_b9qX · 2023-11-01

**Soundness:** 3 good
**Presentation:** 4 excellent
**Contribution:** 2 fair
**Rating:** 3
**Confidence:** 4

**Summary:**

This work proposes a framework called RePLan that integrates multiple foundation model components into a system for iterative high-level robot planning and low-level robot reward code generation. There are five modules in RePLan: 1) LLM for High-level semantic planning to propose intermediate language primitives, 2) VLM for perception for scene state estimation and motion error explanation, 3) LLM for low-level reward code generation given a language primitive and scene state, 4) Motion Controller to translate reward code to robot actions, 5) LLM to verify that 1) and 2) outputs are correct. This system is evaluated on 4 simulated robot manipulation scenarios which require from 1 to 4 subtasks to solve, and is compared against Language2Reward, a method that does not do iterative task decomposition or replanning. The method is compared against ablations which remove specific components; only the full RePLan system is able to achieve non-zero success on all evaluation scenarios.

**Strengths:**

- The paper is well-written and easy to follow
- The motivation of the paper is topical since feedback and adaptive replanning is important for foundation models which may hallucinate or require grounding in physical interactions
- The method does not require additional human input compared to the baseline method Language2Reward (just one human input at the beggining, the rest of the replanning and execution is autonomously completed by the foundation model submodules)

**Weaknesses:**

- The VLM Perceiver is one of the most critical parts of the method, but it is not sufficiently explained. Due to the lack of details, I can only assume how it is utilized based on Algorithm 1, in which case I have some major concerns. Since the VLM is the bottleneck for providing feedback for grounding LLM plans and rewards for future LLM iterations. However, details are not shared about how the Perceiver is used, even though it is mentioned that "The High-Level Planner [is used] to decide what it wants to query from the Perceiver". From Algorithm 1, the VLM needs to be used for two use cases: #1 scene state generation `VLM(image_observation)` and #2 `VLM(image_observation, motion_error, language_instruction)`. However, these seem to be quite challenging tasks to naively query for off-the-shelf VLMs. While modern VLMs are fairly robust at narrow vision task domains like object detection or image captioning for internet images, more extended reasoning (such as failure explanation with multiple input contexts) or domain-specific understanding (like robotics reasoning from vision) is still an open problem.
- The evaluation complexity is very limited and does not justify the claims of a "large-scale and long-horizon kitchen environment": it is only in simulation, with relatively high-level and short-horizon tasks. There are two issues: 1) the granularity of intermediate primitives ("pick up the block") is coarse, 2) the horizon length is short, going only up to 4 subtasks required. Previous works in BEHAVIOR-1K, ALFRED, SayCan have studied robotic reasoning with much longer horizons with similar granularity of intermediate primitives.
- The evaluation has very few trials (3 seeds only), so it is hard to draw confident conclusions about the method's quantitative performance.
- The core claim of the work should clarified. If the contribution is the incorporation of LLM and VLM feedback into high level planning, it needs to be compared/discussed against prior works that ground robot planning with additional foundation model feedback. Specifically, [1] incorporates LLM and VLM feedback for closed-loop environment feedback for a High-Level LLM Planner for robot subtasks. If the contribution is the integration of LLM and VLM feedback into code generation, it needs to compared/discussed with other replanning works from code generation [2] or LLM tool use. If the core contribution is the verifier, this needs to be stated more clearly and more details about this module should be provided. If the contribution is the admittedly impressive integration of existing modules, this should be clarified and more details about the bottlenecks of the system (VLM Perceiver, Verification) should be shared in the main text.

[1] "Inner Monologue: Embodied Reasoning through Planning with Language Models", Huang et al. 2022
[2] "Improving Code Generation by Training with Natural Language Feedback", Chen et al. 2023

**Questions:**

- In general, clarifications to my concerns above will be appreciated.
- Can you explain the VLM Perceiver more? For example: Which VLMs are used? How are they used? Which prompts from the LLM Planner are used? How are the motion errors passed to the VLM? How does the verifier coordinate with the VLM? What is the success rate of the VLM (This is in the context of my concerns in the `Weaknesses` above, where I am doubtful that current VLMs may perform well at extended robot reasoning. Prior works may utilize current VLMs for narrow sub-domains in robotics like object detection or success detection, but RePLan requires textual error explanation "Block is in the way of opening the door" which seems quite difficult in an end-to-end zero-shot VLM).
- Can you clarify the failure reasons in Table 1? For example, does the reward code fail, the high level planner fail, or perceiver fail?

---

> ### Author Response · Authors · 2023-11-18
>
> We appreciate the reviewer’s detailed comments and suggestions. We address the reviewer’s specific concerns below:
>
> **1. Missing information about VLM Perceiver.** We apologize for not providing sufficient details about the VLM Perceiver. We have updated our draft to include all prompts used by the Planners, Perceiver, and Verifier in Appendix B and annotate each line in Algorithm 1 with corresponding prompts. In developing our framework, our experiences completely align with your comments that vanilla, off-the-shelf VLMs are not sufficient to solve these tasks. As a motivating example, we naively ask GPT-4V (which has demonstrated amazing vision-language capabilities) to create a plan for solving Tasks 3 and 6 (the reviewer can find these in Appendix D). In both cases, GPT-4V was unable to identify the correct plan. For Task 3, it was unable to identify that there was block in between the cabinet handles and in Task 4 it could not name the correct cube right off the bat. However, one key insight we find in this work is that we can use an LLM (which has better reasoning skills) to determine what questions it should ask the VLM in a constrained, VQA-style setting in order to update its understanding of the environment. For example, if the Planner asks the  VLM: “Robot in a simulation wants to open the microwave door, can’t. Something else stopping it?” The VLM is able to provide the caption that there is a kettle in front of the door. To prevent overfitting to one question, the LLM asks the VLM multiple iterations of the question (see Figure B9 in Appendix B) and then based on its own knowledge of the world state, it provides a summary answer based on feedback from the VLM of what the most likely explanation. The Planner can also query the Perceiver to update its knowledge on object states, for example: “What is the colour of the crate?” or “Do you see a green apple?” (see Figures B14-B16 in Appendix B). We find that combining LLMs to effectively prompt VLMs and using Verifiers to make sure their outputs are consistent are essential for VLMs to provide useful feedback. We also provide an ablation study in Appendix C2 on the performance of two open-source models and GPT-4V on object recognition and reasoning.
>
> **2. Limited evaluation complexity.** We apologize for the unclear writing and have fixed our description of the number of actions the robot does to accomplish each goal. When we consider an action as any time the robot needs to do something to satisfy an MPC reward or decides to call the VLM to obtain information, most tasks requires on average between 7 and 11 subtasks without human intervention (with some runs using up to 17 depending on how the robot decides to replan). We show a Table in Appendix C (Figure C1) breaking down the number of MPC actions vs VLM actions for each task. This is comparable to the number of actions required by the longest-horizon tasks in SayCan, but those tasks do not have obstacles preventing the robot from following the high-level instructions and do not require replanning based on environment feedback. Furthermore, SayCan utilizes primitive skills to maintain a reliable subtask execution, which we do not assume to have. Not using primitive skills also poses significant challenges to the reliability of the method.
> Few evaluation trials. We have increased the number of trials for each task to 10. We have also added two new environments and three more tasks.
>
> **3. Clarification of core claim.** We apologize for not making the core contributions clear. Our core contribution with this work is as follows: we develop an agent to handle long-term, multi-stage tasks by combining four key aspects: using perceiver models for high-level replanning, creating hierarchical plans with language models, verifying outputs from these language models, and adaptive robot behavior through reward generation. We have updated the introduction to better reflect this. Thank you for pointing out a missing reference to Inner Monologue (IM). While both our works show that robots can perform more complex and longer-horizon tasks by receiving feedback from different sources while executing a task, we are different in the following ways: (a)Their framework uses a VLM at the beginning of the scene for object recognition, but while the scene is progressing, they obtain feedback from humans in some experiments, or an object recognition model to update knowledge about what objects are present/absent in the scene. They do not rely on VLMs for replanning  or providing reasons when the success detector fails. This is most similar to row 3 in our Table 1. Our tasks are not solvable from the presence/absence of objects alone, and our framework is about to solve them without human intervention. (b) When the success detector fails, the robot retries or selects another low-level policy to redo the task (not necessarily to determine reasons why the task failed based on the scene state).

---

> > ### Author Response · Authors · 2023-11-18
> >
> > (continuation from previous comment due to character limit)
> >
> > **4. Failure reasons.** In the updated draft, we include the following additional analyses. In Appendix C1, we test the motion controller module on common motions done in our experiments to ensure it is consistent. We rerun these motions 20 times each. For the errors that occur in our full model framework, we add an analysis of the errors briefly in the Results section in more detail in Appendix C3. The main errors were from the Perceiver-Planner interaction in diagnosing possible failure reasons. In one failed case, the Perceiver did in fact correctly identify the kettle was blocking the microwave in 4/6 questions, but in 1 response it postulated that the cabinet might be the problem. The Planner then replanned for the robot to move the cabinet out of the way instead of the kettle. In another error case for Task 6, the Planner asked the Perceiver “Which cube has the same colour?” which is vague and does not provide enough context, and so the Perceiver did not reply sensibly. In Task 3, the errors were mainly due to MPC failures of the robot trying to remove the red block since it is an intricate action.

---

> > > ### Comment · Reviewer_b9qX · 2023-11-22
> > > **Thanks for the response**
> > >
> > > Thanks for the response. You have clarified some of my concerns, and hope that the main manuscript will be updated to increase the clarity and presentation of your work.
> > >
> > > I will discuss with reviewers in the next phase; I am open to increasing my score to a 4.

---

> > > > ### Author Response · Authors · 2023-11-23
> > > >
> > > > Dear Reviewer b9qX,
> > > >
> > > > Thank you for considering our work and for your openness to increasing the score. To help us refine our work further, can you indicate any particular aspects of the paper that, if improved, would make you more inclined to consider a higher score? We would really appreciate any insights on how we can improve our work or address any remaining concerns. Thank you in advance for your guidance.
> > > >
> > > > Best regards,
> > > > Authors

---

### Official Review · Reviewer_jtHn · 2023-11-10

**Soundness:** 2 fair
**Presentation:** 3 good
**Contribution:** 2 fair
**Rating:** 6
**Confidence:** 4

**Summary:**

This paper presents a way to perform multi-stage planning with LLMs. the method combines multiple LLMs and VLM to reason about long-horizon tasks in a closed-loop fashion. overall the approach is good but I feel it requires more testing on wide variety of tasks to test the generalization of the approach. Furthermore, the testing of the motion planner needs to be more thourough.

**Strengths:**

1. The overall proposed method seems to be interesting and probably could work for a large variety of planning tasks.
2. The tasks are a bit more complex then simple pick and place and illustrate more complex reasoning.

==================
I am increasing the score to 6 after rebuttal discussions.

**Weaknesses:**

1. the authors need to provide more environments or tasks to show the robustness of their method. The authors can use LLMs to generate tasks which are long-horizon to come up with more varieties of task so that the proposed method could be more thoroughly tested. I think this remains to be verified.

**Questions:**

1. How do you test your motion planner module?
2. You say the weakness of your method is the VLM perceiver. Have you tried testing rest of your method by providing text description of the scene by creating a template and keeping the other modules? That can provide more insights of reasoning and control modules of your method?
3. Can provide more results for your system by generating more long-horizon tasks?

---

> ### Author Response · Authors · 2023-11-18
>
> Thank you for your feedback, and we are glad that you find our work interesting. To address your concerns:
>
> **1. More environments and tasks.** We have included two new environments in this iteration with three new tasks. A description can be found in Section 4.1 and the results in Table 1. The environments are a cabinet scene with a lever that unlocks the cabinet (the cabinet is locked when the scene starts). The robot must reason that there are no physical obstacles in the scene and must figure out that pulling the lever unlocks the door in order to accomplish the task. The other environment contains a red crate with two blocks, one red and one yellow. The first task in this environment is to place the cube with the same colour as the crate on the crate, and so the robot must use the Perceiver to determine the correct cube (otherwise it could just get 50% by guessing). The final task is in the same environment; the robot must put the red cube on the crate but there is a yellow cube already on the crate, preventing placement of the red block. The robot must first remove the yellow cube. In addition to integrating these new tasks, we also increased the number of times we ran each of the 7 tasks (from 3 to 10) to further investigate the robustness of our method.
>
> **2. Testing planner module.** For each important motion in our tasks, we first do multiple MPC runs with ground-truth reward functions to test the stability. We pick four of the important motions and include the test results in Appendix C.1 in our updated draft.
>
> **3. VLM Perceiver Weakness.** We thank the reviewer for their suggestion. We have updated our prompting mechanism to alleviate the problems incurred by the VLM perceiver (the reviewer can refer to Appendix B in our updated draft for the detailed prompts, as well as Algorithm 1). As a result, we believe the VLM perceiver is no longer a dominant issue. In Table 1, we demonstrate the impact of only providing a text description of the objects in the scene at start-up (RePLaNn [no Perceiver]), which significantly drops the performance. We include experiments in Appendix C2 demonstrating the object recognition and object reasoning skills of two open-source VLMs and GPT-4V. We find that combining open-source models with segmentation models can also boost object recognition capabilities (Section 4.2 and Appendix C2). We include an additional example of why LLM-constrained VLM prompting is important for our method to work in Appendix D, where we ask GPT-4V to solve Tasks 3 and 6 out-of-the-box, and find that it is unable to. Finally, we include a more in-depth error analysis of the failure cases in Appendix C3 and find that there are still some small issues with the Perceiver-Planner interaction, which could be improved with better prompting. For example, in Task 6, the Planner asks the Perceiver “Which cube has the same colour?” which is very vague, and the Perceiver ends up reply nonsensically and the robot picks the wrong cube.

---

> > ### Author Response · Authors · 2023-11-23
> >
> > Dear Reviewer jtHn,
> >
> > We hope that we have responded to all of your concerns. We welcome any feedback you have in light of our latest revisions, and if you have any additional questions, please let us know!
> >
> > Cheers,
> > Authors

---

### Author Response · Authors · 2023-11-18
**Revision 1**

To all reviewers:

We sincerely thank all reviewers for their thoughtful comments and feedback. Based on the suggestions, we have made major updates to our manuscript, especially in experimental evaluations. We list the changes below.

### Prompts

As requested by reviewers, we have provided details of our prompts in  Appendix B and Algorithm 1. We have updated our prompting mechanism which vastly improves the consistency of our model. As a particular example, in our initial draft, we point out that the VLM perceiver was often not reliable enough to produce diagnoses from a single prompt. However, through our updated prompting (Prompts B.9-12), we believe this has been alleviated due to a better prompting scheme between the Planner and Perceiver and is no longer an issue.

### Evaluations

Multiple reviewers point out the lack of breadth and few trials in our initial evaluations. In light of this, we include two new environments and three new tasks. We also provide quantitative evaluations with  increase the number of trials to 10 each for all  each experiment .

The new environments and tasks are:
- **Wooden cabinet and a lever switch** (Task 5). In this scene, the wooden cabinet is locked via a lever switch. The robot is asked to find the blue cube inside the wooden cabinet. To open the cabinet, the robot needs to perceive and reason about the locking mechanism to figure out the way to activate the lever switch.
- **Colored cubes scene** (Taks 6-7). There are two colored cubes (one yellow and one red) and a colored crate (either yellow or red) in this scene. The goal of this scene is to put the cube that has the same color as the crate on the crate. This environment is designed to evaluate the model’s ability to reason through the visual attributes of the objects, as well as the ability to execute conditional plans (e.g., “If the color of the cube on the left matches the crate, put that cube on the crate”). Two tasks are associated with this scene.
    * *Task 6: The two coloured cubes are placed on opposite sides of the crate.* The robot needs to identify the correct cube and move it on the crate.
    * *Task 7: The cube with the wrong color is initialized on the crate.* The robot will fail to put the correct cube on the crate before removing the wrong one. This task evaluates the model’s ability to diagnose failure through visual attributes reasoning.

**In total, our experiments resulted in a total of 401 MPC action calls and 101 VLM action calls, allowing the robot to execute tasks consisting of up to 17 steps.**

### Draft

We have also made multiple changes to our draft, thanks to the suggestions raised by the reviewers. Specifically, we have added more details in the Results and Appendix on experimental error analysis. We also include ablations of our Perceiver and MPC module. Finally, we include baseline experiment plans from a non-LLM-based method (PDDL) and vanilla GPT-4V (released on Nov 5 2023, after the submission).

The detailed changes in the main text are:

- We have fixed multiple typos, phrasing issues, and formatting issues.
- We have reworked the Introduction and Related Works section to provide a better motivation for our work.
- We have updated Algorithm 1. We fixed a few errors and added links to the exact prompts.
- In the Experiment section, we added the descriptions of the new environments and new tasks. We also updated Sec 4.4 to provide a more thorough results analysis.
- In the Discussion and Limitations section, we updated our comments on the VLM perceiver given our latest evaluations. We also included some discussions on the newly released GPT-4V model.
- We have also provided more details in the Appendix. The detailed additions are:
- In Appendix A, we provide the initial and finished scene images, as well as the text instructions for each of the tasks.
- In Appendix B, we list  the exact prompts that we use in our model and a walk-through. The readers can also cross-reference Algorithm 1.
- In Appendix C, we provide more details on our model, including the distribution of VLM and MPC steps in the 7 tested tasks, additional experiments verifying the motion planning module, an ablation study for different VLM models, and finally the error case analyses of our model. We also test 3 state-of-the-art VLMs, including GPT-4V.
- In Appendix D, we provide an additional experiment to test the ad-hoc planning ability of GPT-4V.
- In Appendix E, we provide a comparison between our model and TAMP.

---

### Author Response · Authors · 2023-11-23
**Revision 2**

Dear Reviewers,

In light of the earlier feedback and supplementary experiments conducted, the subsequent table provides a summary of our results:


**Table 1: Number of times that the models completed Tasks 1-7 out of 10 runs. Average completion rates across 7 tasks are listed in the last column.**

| Model                        | Task 1  | Task 2  | Task 3 | Task 4 | Task 5 | Task 6 | Task 7 | Average  |
|------------------------------|---------|---------|--------|--------|--------|--------|--------|----------|
| Our Model [full]             | **100%**| **100%**| **60%**| **80%**| **100%**| 90%    | **90%** | **88.6%**|
| Our Model [no Verifier]      | 80%     | 80%     | 20%    | 60%    | **100%**| **100%**| 80%     | 74.3%    |
| Our Model [no Perceiver]     | 90%     | 90%     | 30%    | 20%    | 50%    | 20%    | 0%     | 42.9%    |
| Our Model [no Replan]        | 80%     | 70%     | 0%     | 0%     | 0%     | 80%    | 10%    | 34.3%    |
| Language to Rewards [1]      | 90%     | 20%     | 0%     | 0%     | 0%     | 50%    | 10%    | 24.3%    |

The videos of our experiments are available at our website:
https://sites.google.com/view/replan-iclr/home


## Long-Horizon Tasks

In addition to the seven tasks in the table above, we introduce an additional long-horizon task here.

**Task 8 (Figure A.1o, A.1p, video available in our website)** In this task, the robot is asked to open a slide cabinet, which is locked via a weight sensor. The weight is hidden in another cabinet, with a microwave as a distraction option. The robot needs to first search for the weight by opening the microwave and the cabinet, then retrieve the weight out of the cabinet, put the weight onto the weight sensor and eventually unlock and open the slide cabinet.

**Result for Task 8** RePLan achieves **70%** completion rate out of 10 runs while Language to Rewards completes 0%.

[1] Wenhao Yu et al, “Language to Rewards for Robotic Skill Synthesis”. (2023)

---

### Meta-Review · Area_Chair_kvaK · 2023-12-05

**Metareview:**

**Summary**: The paper proposes RePLan, a framework for iterative high-level robot planning and low-level robot reward code generation. The system has 5 components: 1. an LLM for high-level semantic planning to chain together language primitives; 2. a VLM for perception for state estimation; 3. an LLM for low-level reward code generation given the language primitive and perceived state; 4. a motion controller than converts reward code to actions; 5. an LLM that verifies the individual components' correctness. RePLan is evaluated on 4 manipulation scenarios in simulation and compared to Language2Reward which does not do iterative decomposition or replanning. Ablation results showcase that all components are necessary for successful evaluation.

**Strengths**:
- The paper is well-written and easy to follow.
- Comprehensive related work.

**Weaknesses**:
- Limited evaluation complexity: the method is evaluated only in simulation, on relatively high-level and short-horizon tasks (horizon length goes only up to 4 subtasks). Since the method is a relative of papers like BEHAVIOR-1K, ALFRED, SayCan, I believe appropriate evaluation should raise to their level and have longer horizon tasks and real-world deployment. Additionally, the tasks themselves are relatively narrow and share some similarities amongst each other.
- Unclear contribution/novelty: The contribution when compared to much prior work in this space is not clear. The related work is extensive, but is presented more like a laundry list of recent works.

Overall, most reviewers agree that the evaluation is much too limited for publication at ICLR. I suggest that the authors expand their evaluation tasks and environments, and add some real world demos as well.

**Justification For Why Not Higher Score:**

Very limited evaluation which all reviewers acknowledge, including the one reviewer that gave a 6.

**Justification For Why Not Lower Score:**

N/A

---

### Decision · Program_Chairs · 2024-01-16

Reject